# Cyclin-dependent kinase control of motile ciliogenesis

Eszter K Vladar[1,2,3]*, Miranda B Stratton[4], Maxwell L Saal[2,3], Glicella Salazar-De Simone[5], Xiangyuan Wang[6], Debra Wolgemuth[6], Tim Stearns[4,7], Jeffrey D Axelrod[1]

[1]Department of Pathology, Stanford University School of Medicine, Stanford, United States; [2]Division of Pulmonary Sciences and Critical Care Medicine, Department of Medicine, University of Colorado School of Medicine, Aurora, United States; [3]Department of Cell and Developmental Biology, University of Colorado School of Medicine, Aurora, United States; [4]Department of Biology, Stanford University, Stanford, United States; [5]Center for Radiological Research, Columbia University Medical Center, New York, United States; [6]Department of Genetics & Development, Columbia University Medical Center, New York, United States; [7]Department of Genetics, Stanford University School of Medicine, Stanford, United States

**Abstract** Cycling cells maintain centriole number at precisely two per cell in part by limiting their duplication to S phase under the control of the cell cycle machinery. In contrast, postmitotic multiciliated cells (MCCs) uncouple centriole assembly from cell cycle progression and produce hundreds of centrioles in the absence of DNA replication to serve as basal bodies for motile cilia. Although some cell cycle regulators have previously been implicated in motile ciliogenesis, how the cell cycle machinery is employed to amplify centrioles is unclear. We use transgenic mice and primary airway epithelial cell culture to show that Cdk2, the kinase responsible for the G1 to S phase transition, is also required in MCCs to initiate motile ciliogenesis. While Cdk2 is coupled with cyclins E and A2 during cell division, cyclin A1 is required during ciliogenesis, contributing to an alternative regulatory landscape that facilitates centriole amplification without DNA replication.
DOI: https://doi.org/10.7554/eLife.36375.001

*For correspondence:
eszter.vladar@ucdenver.edu

**Competing interests:** The authors declare that no competing interests exist.

## Introduction

Centrioles are microtubule-based, radially symmetric, cylindrical structures. A pair of centrioles, and surrounding pericentriolar material, comprise the centrosome (*Vertii et al., 2016*). In interphase, one of the centrioles can serve as basal body for a primary cilium while the pericentriolar material is a principal site for cytoplasmic microtubule nucleation and organization. In mitosis, centrosomes regulate the assembly and orientation of the mitotic spindle. In these roles, centrioles are vital to fundamental cellular processes including signal transduction, intracellular trafficking and cell division. Dysfunction of centrioles or associated structures results in developmental and adult tissue maintenance defects (*Bettencourt-Dias et al., 2011*), which have been linked to human disease, notably ciliopathies and cancer.

In dividing cells, centriole generation occurs in S phase of the cell cycle when exactly two new (daughter) centrioles assemble, each next to an existing (mother) centriole (*Figure 1—figure supplement 1A*) (*Fırat-Karalar and Stearns, 2014*). The resulting two centrosomes then segregate into the two daughter cells during mitosis to reduce the centriole number to two in each daughter. Control of centriole number is important for proper cellular function: extra centrioles can lead to abnormal

mitoses (*Yang et al., 2008*) or give rise to extra primary cilia that results in defective signaling (*Mahjoub and Stearns, 2012*). Centriole generation is highly regulated to ensure the assembly of the correct number of structures by limiting duplication to S phase, by limiting assembly to one new daughter centriole alongside each mother centriole, by blocking the de novo (noncentriolar) generation of extra centrioles, and by coupling centriole and DNA duplication under common timing and regulation.

Multiciliated cells (MCCs) of the airway, ependymal, middle ear and oviduct epithelia break the rules that govern centriole formation in dividing cells, as they assemble, depending on cell type, dozens to hundreds of centrioles in a single postmitotic cytoplasm to act as basal bodies to motile cilia (*Figure 1—figure supplement 1B*) (*Meunier and Azimzadeh, 2016*). MCC fate is acquired in a Notch signaling-dependent manner, with cells experiencing Notch activation becoming secretory cells and cells not experiencing Notch activation progressing down the MCC pathway (*Tsao et al., 2009*). Ciliogenesis is initiated when nascent MCCs launch a MCC-specific transcriptional program to express hundreds of ciliary genes. Next, centrioles form in the cytoplasm, traffic to and dock with the apical plasma membrane and elongate a motile ciliary axoneme. In contrast to dividing cells, MCCs are capable of (1) generating centrioles in the postmitotic or G0 phase, (2) generating many daughter centrioles per mother centriole, (3) using unique structures termed deuterosomes for the de novo (noncentriolar) assembly of centrioles, and (4) uncoupling centriole and DNA duplication. Yet, both MCCs and dividing cells produce apparently structurally identical centrioles, and the MCC pathway appears to rely on many known cell cycle regulated centrosome duplication factors such as the Plk4 kinase and structural components including Sass6 and Centrin proteins (*Vladar and Stearns, 2007*). This suggests that there are both universal pathways as well as MCC specific alterations that permit large scale postmitotic centriole amplification.

Mechanisms that control centriole number in dividing cells consist of both centriole-intrinsic and cytoplasmic events. Limiting centriole duplication to S phase is ultimately under the control of the cell cycle machinery. Timely entry and progression through S phase is regulated by Cyclin-dependent kinase 2 (Cdk2) complexed with cyclins E or A2 (*Hochegger et al., 2008*). Distinct Cdk-cyclin pairs control cell cycle transitions through a highly orchestrated program of Cdk posttranslational modification and cyclin expression and degradation (*Heim et al., 2017*). The G1 to S phase transition occurs when mitogenic stimulation leads to activation of the Cdk4/6-cyclin D complex, which acts to dissociate the E2F1 transcription factor from the Dp1 and Retinoblastoma (Rb) proteins, leading to E2F1 activation. E2F1 turns on key S phase genes, notably cyclins E and A2 and DNA synthesis factors. Cyclin E binding leads to activation of Cdk2. Then Cdk2, coupled to cyclin A2 (which further phosphorylates E2F1) initiates entry into and controls progression through S phase and the twin events of centriole and DNA duplication (*Hinchcliffe et al., 1999*; *Lacey et al., 1999*; *Matsumoto et al., 1999*). The precise events that link the cell cycle machinery to centriole duplication are not yet clear, however, many regulators localize to the centrosome (*Kodani et al., 2015*) and may control Plk4 stability and activity (*Korzeniewski et al., 2009*).

Interestingly, recent insights into MCC differentiation revealed that multiple cell cycle regulators also play important roles in motile ciliogenesis (*Meunier and Azimzadeh, 2016*). The initiation of the MCC gene expression program depends on a transcriptional complex (EMD complex) comprising the E2F4 or E2F5 transcription factor, a Geminin family transcriptional activator (Mcidas or Gmnc) and Dp1 (*Ma et al., 2014*). This complex is highly similar in make up to the E2F1/Rb/Dp1 complex that controls G1 to S phase progression, which suggests the existence of universal cell cycle regulatory mechanisms to create a permissive environment for centriole assembly. Downstream of EMD, the Myb and p73 transcription factors, also well known for their cell cycle functions (*Allocati et al., 2012*), turn on further MCC genes (*Tan et al., 2013*; *Marshall et al., 2016*). Finally, the G2 to M phase cell cycle machinery, including Cdk1 was shown regulate intermediate stages of ciliogenesis in ependymal MCCs (*Al Jord et al., 2017*).

To investigate the extent to which the G1 to S phase cell cycle machinery is involved in motile ciliogenesis, we tested the role of Cdk2 using primary mouse tracheal epithelial cell (MTEC) culture, which recapitulates basal (stem) cell proliferation and subsequent MCC differentiation (*You et al., 2002*). Here, we show that downstream of MCC fate acquisition, Cdk2 is required to initiate and maintain the MCC gene expression program during ciliogenesis. Consistently, MCCs driven to differentiate in the absence of Cdk2 activity fail to undergo ciliogenesis. Unlike in dividing cells, Cdk2 appears to function during ciliogenesis together with Cyclin A1 (Ccna1), a cognate cyclin thought to

activate Cdk2 in meiotic cell cycles (*Joshi et al., 2009*). In sum, our results indicate that Cdk2 activity is universal regulator of centriole assembly, and the involvement of Ccna1, a noncanonical binding partner in somatic cells, and likely other factors enable it to function in a postmitotic cell and drive centriole amplification.

## Results

### Cyclin-dependent kinase activity is required for motile ciliogenesis

We used the MTEC culture system to test the requirement for Cdk activity during MCC differentiation (*Figure 1—figure supplement 1A–C*). MTECs are a faithful model of airway epithelial development and regeneration and permit the observation and manipulation of both the proliferative and differentiation phases of the process (*You et al., 2002*). At maturity, MTECs contain all the cell types of the donor tissue, including MCCs, secretory cells and basal stem cells. The cultures are initiated by seeding basal stem cells isolated from mouse trachea onto porous Transwell membranes (*Vladar and Brody, 2013*). Cells proliferate to confluence while submerged in medium, then are lifted to an air-liquid interface (ALI) which promotes the differentiation of airway epithelial cells (*Figure 1—figure supplement 1C*). Culture progression follows a timeline in which basal cells proliferate during days 1–3, the resulting confluent cell layer acquires a columnar, apically compacted morphology from days 3–5, ALI is created on day 5 (ALI + 0 days), and MTECs are mature by ALI + 14 d with MCCs and secretory cells at the luminal surface with underlying basal cells. MCC fate acquisition and motile ciliogenesis occur asynchronously in both the in vivo airway epithelium and in MTECs (*Vladar and Stearns, 2007*), but early ALI cultures are strongly enriched for MCCs in the initial stages of motile ciliogenesis.

To test the requirement for Cdk activity, we treated differentiating MTECs with well-characterized, dose-dependent small molecule inhibitors that predominantly act on Cdk1 and Cdk2 (Cdkis). We verified the previously established ability of these Cdkis to produce cell cycle arrest in 293T/17 cells before using in MTECs (not shown). MTECs were treated from ALI + 0 to ALI + 4 d (chronic treatment) with the Cdkis Purvalanol A, Roscovitine, NU6140 and Cdk2 Inhibitor III, then labeled at ALI + 4 d with anti-acetylated α-Tubulin (ac. α-Tub) antibody to mark cilia. While untreated cultures contained many MCCs, we found that all four Cdkis blocked ciliogenesis (*Figure 1—figure supplement 2*). To characterize this phenotype, we labeled cells with antibodies against structural and regulatory components of the ciliogenesis pathway, and found that Cdki treated MTECs failed to form centrioles (ac. α-Tub, Odf2, Pericentrin, Sass6, and Plk4), deuterosomes (Ccdc67, also known as Deup1) or ciliary axonemes (ac. α-Tub), indicating an early ciliogenesis arrest (*Figure 1A–B*). Motile ciliogenesis is initiated and maintained by the expression of ciliary genes via the sequential activity of MCC transcription factors (MCC TFs, see *Figure 1—figure supplement 1B*) (*Meunier and Azimzadeh, 2016*). We found that Cdki treated cells did not express the MCC TFs Foxj1 and Myb (*Figure 1A*). Using qRT-PCR, untreated cells showed strong upregulation of both MCC TFs and ciliary genes at ALI + 4 d compared to confluent, but not yet differentiating cells (ALI-1d or day 4 of culture), whereas Cdki treated cells had no detectable levels of MCC TFs and ciliary genes were not upregulated (*Figure 1C*). Cdki treatment therefore blocks motile ciliogenesis at an early step of the pathway at the level of MCC gene expression.

Ciliogenesis arrest was fully reversible for all Cdkis, as MTECs released from ALI + 0 to +4 d Cdki treatment were robustly ciliated by ALI + 8 d (*Figure 1B,D* for NU6140 and Purvalanol A, others not shown). Cdki treatment had no effect on overall epithelial morphology as judged by E-cadherin antibody labeling of apical cell-cell junctions or on the presence of primary cilia (*Figure 1A*). These results suggest that the ciliogenesis arrest is a specific inhibition of the motile ciliogenesis pathway and not a nonspecific detrimental effect on the differentiation or overall health and integrity of the MTECs.

The Cdkis used are known to have dose-dependent activity on multiple Cdk-cyclin complexes, and at higher doses they can also inhibit unrelated kinases (*Knockaert et al., 2002*; *Peyressatre et al., 2015*). We found the while the Cdk1/2-specific Purvalanol A and Roscovitine and the Cdk2-specific NU6140 and Cdk2 Inhibitor III compounds robustly inhibited ciliogenesis, the Cdk1 inhibitor RO3306 had no effect (*Figure 1B* and *Figure 1—figure supplement 2*). Moreover, lentiviral expression of a dominant negative HA-Cdk2 (D145N) construct also blocked ciliogenesis,

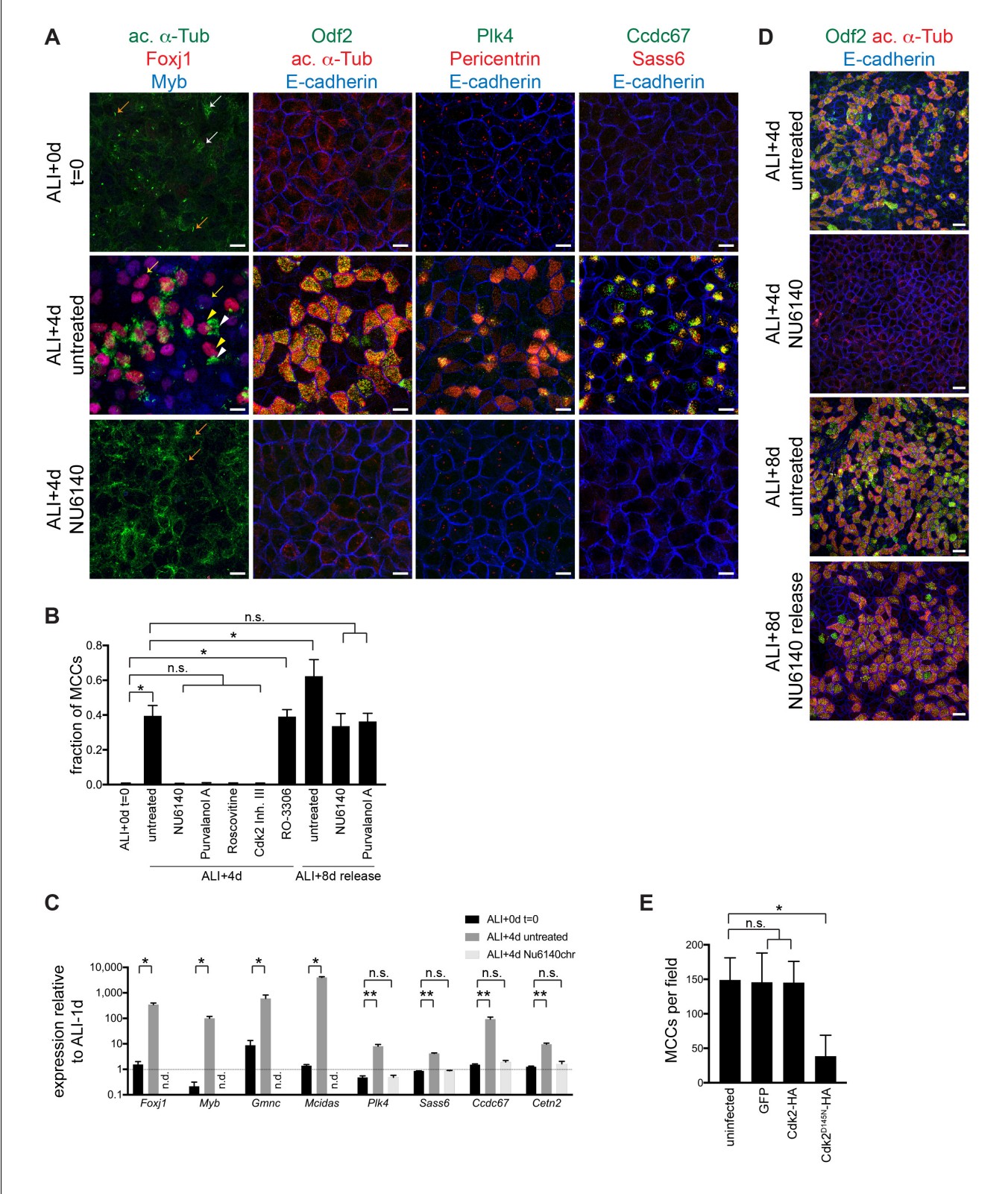

**Figure 1.** Chronic Cdk inhibitor treatment blocks motile ciliogenesis. (**A**) MTECs were treated with NU6140 from ALI + 0 to 4d. They were fixed at ALI + 0 and +4 d and labeled with antibodies to monitor ciliogenesis: left, ac. α-Tub (green), Foxj1 (red) and Myb (blue); center left, Odf2 (green), ac. α-Tub (red) and E-cadherin (blue); center right, Plk4 (green), Pericentrin (red) and E-cadherin (blue); and right, Ccdc67 (green), Sass6 (red) and E-cadherin (blue). MTECs are confluent without any sign of motile ciliogenesis at ALI + 0 d. Untreated ALI + 4 d cells are robustly ciliating, but NU6140 treatment

*Figure 1 continued on next page*

*Figure 1 continued*

blocks all signs of motile ciliogenesis. Ac. α-Tub marks cytoplasmic microtubules in non-MCCs (white arrow) and motile ciliary axonemal tufts in MCCs (white arrowhead); when MCCs are present, the much weaker cytoplasmic signal is not discernible. Foxj1 and Myb mark both ciliating (yellow arrow) MCCs without ac. α-Tub + axonemes and Foxj1 marks mature MCCs (yellow arrowheads) with axonemes. NU6140 treatment has no effect on primary cilium formation (orange arrow) or on apical cell junctions. Scale bar, 10 µm. (B) Quantitation of the Cdki block and the release for Cdki treatment. MCCs were identified by ac. α-Tub labeling. n.s., not significant; *p<0.0001 (C) Realtime PCR results show that the expression of MCC TFs (*Foxj1*, *Myb*, *Gmnc* and *Mcidas*) is suppressed and the expression of ciliary components (*Plk4*, *Sass6*, *Ccdc67* and *Cetn2*) is not upregulated in cells treated with NU6140 from ALI + 0 to 4d. Levels were normalized to *Gapdh* expression and compared to values obtained for MTECs at ALI-1d (n.d. = none detected). n.s., not significant, *p<0.05, **p<0.0001 (D) MTECs were treated with NU6140 from ALI + 0 to 4d, then cultured without Nu6140 until ALI + 8 d. Cells were fixed at ALI + 4 and+8 d and labeled with Odf2 (green), ac. α-Tub (red) and E-cadherin (blue) antibodies to show that MTECs ciliate robustly after release from Cdki treatment. Scale bar, 20 µm. (E) MCCs were quantitated based on ac. α-Tub labeling in MTECs infected with GFP, Cdk2-HA or Cdk2$^{D145N}$-HA lentivirus. Cdk2$^{D145N}$, but not wildtype Cdk2 expression blocks ciliogenesis. Ectopic wildtype Cdk2 expression in MTECs is not sufficient to drive motile ciliogenesis. n.s., not significant; *p<0.000.

DOI: https://doi.org/10.7554/eLife.36375.002

The following figure supplements are available for figure 1:

**Figure supplement 1.** The motile ciliogenesis pathway and the MTEC culture system.
DOI: https://doi.org/10.7554/eLife.36375.003
**Figure supplement 2.** Cdk inhibitor treatment blocks MCC differentiation in MTECs.
DOI: https://doi.org/10.7554/eLife.36375.004

while the expression of wildtype HA-Cdk2 or GFP had no effect (*Figure 1E*). Thus, we conclude that Cdk2 is required for initiation of the motile ciliogenesis pathway, consistent with its previously described role in centriole assembly in S phase.

## Cdk2 acts downstream of MCC specification and upstream of the MCC gene expression program

Upon airway epithelialization, MCC and secretory cell fates are acquired in a Notch signaling-dependent mechanism. Precursor cells in which Notch signaling is activated are diverted to the secretory cell fate, whereas cells in which Notch is not activated continue toward MCC differentiation (*Tsao et al., 2009*) *Figure 1—figure supplement 1A–B*). In the prospective MCCs (those that avoid Notch activation), MCC differentiation may be a default occurrence, though we cannot rule out that some additional external initiating event is required. Subsequent to this cell fate decision point, cells remaining in the MCC pathway activate the MCC gene expression program to turn on ciliary genes (*Brooks and Wallingford, 2014*; *Meunier and Azimzadeh, 2016*). To test the relationship between Cdk2 and the Notch signaling event, we treated MTECs from ALI + 0 to 4d with NU6140 to block Cdk activity and with the γ-secretase inhibitor DAPT to block Notch signaling (*Stubbs et al., 2012*). We monitored ciliogenesis using antibody labeling for ac. α-Tub to mark cilia and Foxj1 to mark MCCs at earlier stages of ciliogenesis (nascent MCCs without mature cilia already express Foxj1) (*You et al., 2004*). As expected, NU6140 treated MTECs lacked MCCs. MTECs treated with DAPT alone had a two-fold increase in MCCs compared to untreated cells as more cells evaded Notch activation and remained in the MCC pathway (*Figure 2A*). However, DAPT treatment was not able to induce MCCs in the presence of NU6140 (*Figure 2A*), indicating that even cells directed toward the MCC fate by Notch inhibition still require Cdk2 activation to continue this progression. Because the Notch decision point occurs before Cdk2 activation, we refer to Cdk2 activation as acting 'downstream' of the Notch decision, although strictly speaking, it is peculiar to argue that Cdk2 activation acts downstream of something that does not occur (Notch signaling).

TRRAP, a component of multiple histone acetyltransferase complexes was recently shown to be required for the initiation of the MCC gene expression program 'downstream' of the Notch signaling event (downstream in the same sense as described for Cdk2 above) (*Wang et al., 2018*). TRRAP nuclear expression arises in prospective MCCs prior to Foxj1 expression. We asked whether TRRAP might act downstream of Cdk2 by assessing TRRAP expression in Cdki treated MTECs. We found that while Nu6140 treated MTECs lacked Foxj1+ MCCs, they contained as many TRRAP+ cells as untreated MTECs at the same stage (*Figure 2B*). This indicates that in prospective MCCs, TRRAP does not act downstream of Cdk2 activation. Cdk2 therefore acts either downstream of or in parallel to TRRAP in the initiation of the MCC gene expression program (*Figure 2C*). Furthermore, the

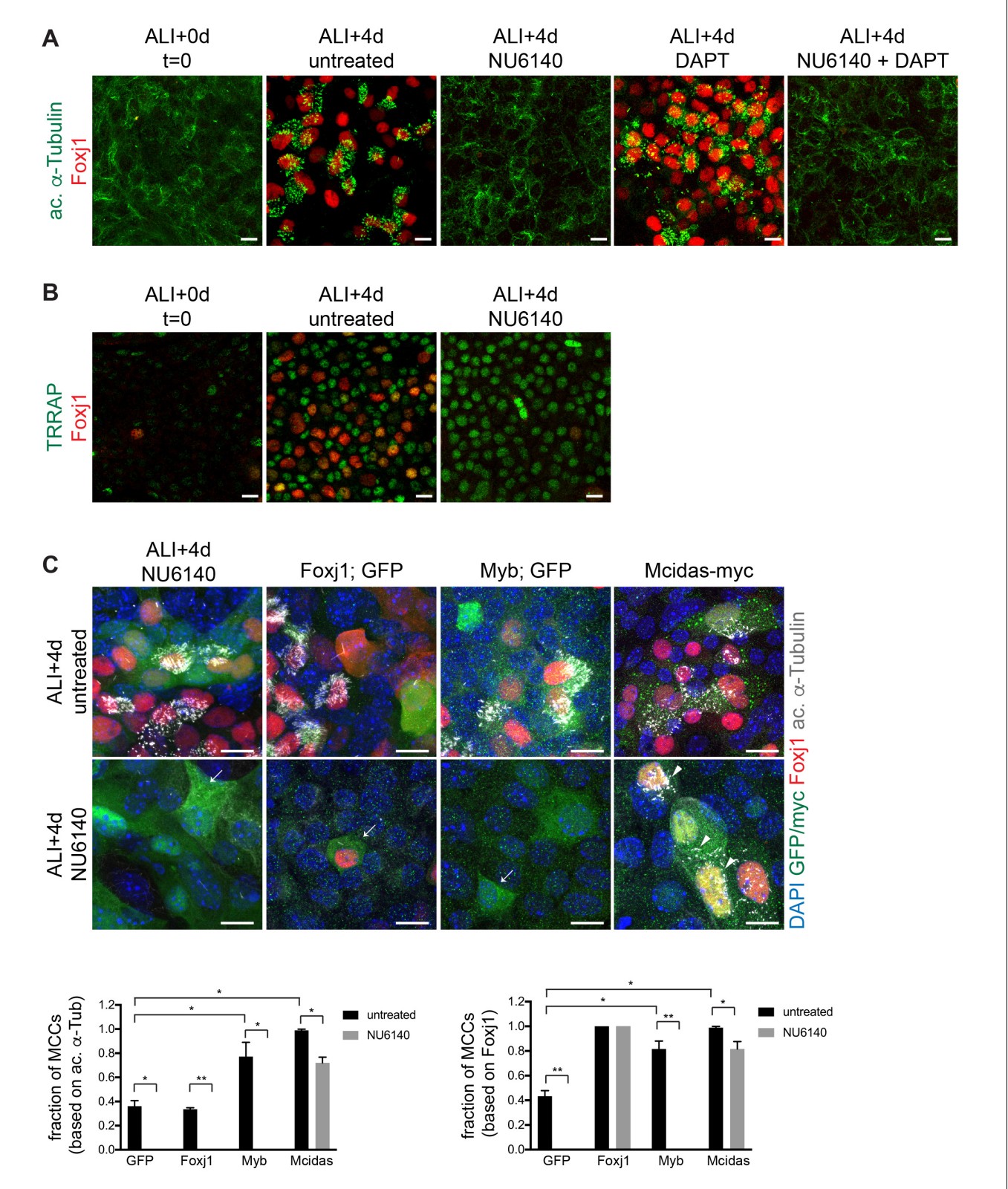

**Figure 2.** Cdk2 acts downstream of Notch signaling and upstream of the MCC gene expression program. (**A**) MTECs were treated with NU6140 and/or DAPT from ALI + 0 to 4d. They were fixed at ALI + 0 and +4 d and labeled with ac. α-Tub (green) and Foxj1 (red) antibodies. DAPT induces MCC formation but not in the presence of NU6140. Scale bar, 10 μm. (**B**) MTECs were treated with NU6140 from ALI + 0 to 4d. They were fixed at ALI + 0 and +4 d and labeled with TRRAP (green) and Foxj1 (red) antibodies. Scale bar, 20 μm. (**C**) MTECs were infected with lentivirus encoding GFP, or Foxj1

*Figure 2 continued on next page*

*Figure 2 continued*

or Myb and GFP from separate promoters, or myc-tagged Mcidas at ALI-2d, then treated with NU6140 from ALI + 0 to 4d. They were fixed at ALI + 4 d and labeled with GFP or myc (green), Foxj1 (red) and ac. α-Tub (white) antibodies and stained with DAPI (blue) to mark nuclei. Only Mcidas, but not GFP, Foxj1 or Myb expression can drive the complete motile ciliogenesis pathway (arrows indicate GFP+ cells without ac. α-Tub+ cilia, arrowheads point to mature myc+ MCCs) in NU6140 treated MTECs. Foxj1 expression leads to nuclear Foxj1 accumulation, but not ac. α-Tub+ cilia. MCCs were quantitated based on Foxj1 and ac. α-Tub expression. Scale bar, 10 μm. *p<0.001, **p<0.0001.

DOI: https://doi.org/10.7554/eLife.36375.005

appearance of TRRAP expression in the absence of Cdk2 activity indicates the adoption of the MCC cell fate, thus separating Cdk2 activity from the MCC cell fate decision.

Next, we more closely investigated the relationship between Cdk2 and the MCC transcriptional regulators Mcidas, Myb and Foxj1. While the precise hierarchy remains unclear, Mcidas (as part of the EMD complex) is known act upstream of the other two, but there is also evidence for feedback regulation and overlap in targets (*Brooks and Wallingford, 2014*). As MCC TFs are repressed by Cdki treatment (*Figure 1A*), we asked if they can function downstream of Cdk2 to drive ciliogenesis when expressed ectopically in the presence of Cdki treatment. MTECs infected at d3 of culture with lentivirus containing the MCC TF Mcidas, then treated with NU6140 from ALI + 0 to 4d were able to carry out the complete motile ciliogenesis pathway as judged by Foxj1 expression and the presence of cilia (*Figure 2C*). Cultures expressing Myb or Foxj1 under these conditions did not undergo ciliogenesis. NU6140 treated cells expressing Foxj1 had nuclear or nucleo-cytoplasmic Foxj1 signal, but never made cilia (*Figure 2C*). This suggests that Cdk2 acts upstream of Mcidas and raises the possibility that Mcidas or one of its binding partners may be a target of Cdk2. Consistent with previous reports, the expression of Mcidas and Myb (*Stubbs et al., 2012*; *Tan et al., 2013*), but not Foxj1 (*You et al., 2004*), was sufficient to drive motile ciliogenesis in untreated MTECs (*Figure 2C*).

Our results place the Cdk2 requirement downstream of the Notch-dependent cell fate decision and upstream of EMD in the MCC gene expression pathway for initiating ciliogenesis. The clear temporal separation of the proliferation (preALI) and differentiation (beginning at ALI + 0 d) phases of the MTEC culture system already made it unlikely that the ciliogenesis block by Cdki treatment starting at ALI + 0 d stems from the disruption of a proliferative event. The fact that Cdk2 acts downstream of the Notch signaling event (*Figure 2A*) indicates that Cdki blocked cells have exited the cell cycle and undergone MCC fate selection, reinforcing the conclusion drawn from TRRAP expression (*Figure 2B*) in these cells. Identifying Cdk2 as an upstream regulator of the MCC transcriptional program fills an important gap in knowledge for this process and further cements a universal role for the cell cycle machinery in centriole generation.

## Cdk2 is required to sustain the MCC gene expression program

Although understanding of the regulatory hierarchy of the program is still emerging, MCC gene expression appears to be a protracted, multi-step process with early and later transcriptional steps driving the sequential execution of cilium biogenesis events (*Meunier and Azimzadeh, 2016*). Results presented thus far indicate that Cdk2 activates MCC gene expression. To test if it is also required for the sustained expression of ciliary genes, we treated MTECs after the onset of ciliogenesis, from ALI + 3 to ALI + 4 d, with NU6140 (acute treatment). Acute Cdki treatment resulted in an immediate ciliogenesis arrest (*Figure 3A–B*), indicating that Cdk2 is not only required to initiate, but also to sustain the pathway. To characterize the Cdki arrest, we used centriole markers (γ-Tubulin and Odf2) to quantitate the fraction of MCCs at different stages of ciliogenesis (*Figure 1—figure supplement 1B*), as previously described (*Vladar and Stearns, 2007*). Untreated cultures consistently transition from a heterogeneous population of ciliating cells with some mature MCCs to a population with fewer ciliating and more mature MCCs between ALI + 3 to 4d (*Figure 3A–B*). We found that Cdki treatment blocked this transition and cells arrested at all stages of ciliogenesis (*Figure 3A–B*). The MCC TF *Myb* is expressed first during motile ciliogenesis; then it turns on *Foxj1* (along with other TFs) and is then downregulated while *Foxj1* stays on (*Tan et al., 2013*). Thus, the relative expression of these two MCC TFs can be used to monitor ciliogenesis progression. In untreated MTECs, we consistently detected a decrease in Myb+/Foxj1- and the increase in Myb-/Foxj1 +cells from ALI + 3 to 4d by antibody labeling (*Figure 3A,C*). However, we did not detect a statistically significant decrease in the Myb+/Foxj1- population and the Myb-/Foxj1+ population

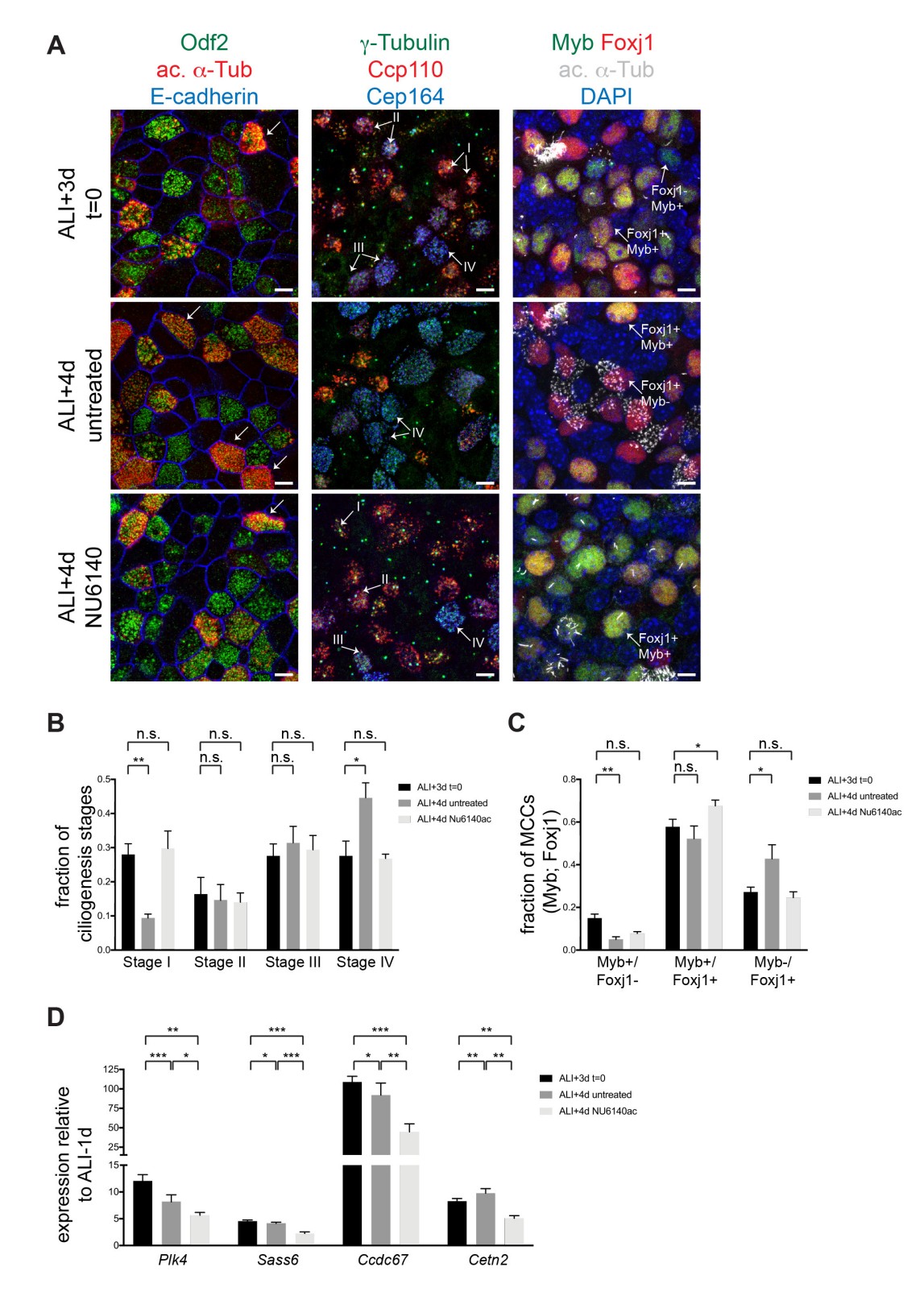

**Figure 3.** Cdk2 activity is required to sustain motile ciliogenesis. (**A**) MTECs were treated with NU6140 from ALI + 3 to 4d. They were fixed at ALI + 3 and+4 d and labeled with antibodies to monitor ciliogenesis: left, Odf2 (green), ac. α-Tub (red) and E-cadherin (blue); center, γ-Tubulin (green) Ccp110 (red) and Cep164 (blue); and right, DAPI (blue), Myb (green), Foxj1 (red) and ac. α-Tub (white). MTECs at ALI + 3 d contain predominantly ciliating cells. Many mature MCCs emerge in untreated ALI + 4 d MTECs, but not in the presence of NU6140. Left panels: Stage IV MCCs are marked by ac. α-

*Figure 3 continued on next page*

*Figure 3 continued*

Tub + cilia (arrow). Center panels: Stage I MCCs are Ccp110+, Cep164-; Stage II MCCs are Ccp110+, Cep164+; Stage III MCCs are Ccp110 low, Cep164 high and Stage IV MCCs are Ccp110-, Cep164+. Right panels: Stage I MCCs are Foxj1-, Myb+; Stage II MCCs are Foxj1+, Myb+; and Stage III-IV MCCs are Foxj1+, Myb-. Scale bar, 10 µm. n.s., not significant; *p<0.01, **p<0.001 (B) Quantitation of MCCs at Stages I-IV (see *Figure 1—figure supplement 1B*) with and without acute NU6140 treatment. n.s., not significant; *p<0.05 (C) Quantitation of Foxj1 and Myb positive cells with and without acute NU6140 treatment. n.s., not significant; *p<0.05, **p<0.01 (D) Realtime PCR showing that the expression of ciliary components *Plk4*, *Sass6*, *Ccdc67* and *Cetn2* decreases in cells treated with NU6140 from ALI + 3 to 4d. Note that as ciliogenesis peaks expression levels for some components also decrease in untreated MTECs from ALI + 3 to 4d. Levels were normalized to *Gapdh* expression and compared to values obtained for MTECs at ALI-1d. *p<0.05; **p<0.01; ***p<0.001.

DOI: https://doi.org/10.7554/eLife.36375.006

The following figure supplement is available for figure 3:

**Figure supplement 1.** NU6140 treatment does not disrupt cilia on mature MCCs.

DOI: https://doi.org/10.7554/eLife.36375.007

failed to increase under Cdki treatment (*Figure 3C*), indicating that cells failed to progress in the ciliogenesis pathway. We also found that the expression of structural and regulatory ciliary genes (*Ccdc67*, *Plk4*, *Sass6* and *Cetn2*) declined upon Cdki treatment, which suggests that continued Cdk2 activity is required to maintain MCC gene expression (*Figure 3D*). We do not distinguish whether ciliogenesis arrests due to impairment of a signal or due to the depletion of ciliogenic components, although arrest is evident at the level of the transcriptional network. We again rule out that the NU6140 arrest is simply nonspecific injury to MCCs as treatment had no effect on mature MCCs (*Figure 3—figure supplement 1*).

## Cdk2 is active and localizes to the nucleus and centrioles during motile ciliogenesis

To ensure the timely execution of S phase events, Cdk2 activation in dividing cells is under tight control by nucleo-cytoplasmic shuttling, posttranslational modification and cyclin binding. Although these regulatory events may not all be at play in a postmitotic cell, we sought evidence that Cdk2 is active in ciliating MCCs. Using lentivirally expressed wildtype HA-Cdk2, we observed that Cdk2 was enriched in the nucleus in ciliating cells (nuclear to cytoplasmic ratio = 1.96±0.26), but not in mature (nuclear to cytoplasmic ratio = 0.74±0.07) MCCs (*Figure 4A–B*). Although total Cdk2 levels did not change during ciliogenesis at the protein or transcript level, we detected a peak in Cdk2 bearing the activating Thr160 phosphorylation (*Gu et al., 1992*) by Western blot at ALI+2 to +4d, a time interval that is enriched for MCCs in early ciliogenesis (*Figure 4C*, *Figure 4—figure supplement 1A–B* and *Figure 1—figure supplement 1C*). The presence of nuclear Cdk2 in ciliating MCCs and the enrichment of phospho-Thr160 Cdk2 in ciliating MTECs indicates that Cdk2 is active during early ciliogenesis. These signs of Cdk2 activation, together with the requirement for Cdk2 activity to sustain MCC TF expression support a key role for Cdk2 in the initiation and maintenance of the motile ciliogenesis program. Interestingly, we found that HA-Cdk2 also localized to centrioles in both ciliating and mature MCCs (*Figure 4D*). As we were not able to observe endogenous Cdk2 at MCC centrioles due to the lack of effective antibodies, we could not confirm that this centriolar localization of ectopically expressed Cdk2 is representative of endogenous Cdk2 distribution. However, others have reported that Cdk2 can localize to the centrosome in cycling cells (*Kodani et al., 2015*), and that it has centrosomal phosphotargets (*Chen et al., 2002*; *Okuda et al., 2000*). Thus, both the nuclear and centriolar pools of Cdk2 may be involved in ciliogenesis regulation.

## Ccna1 is upregulated during motile ciliogenesis

A- and E-type cyclins associate with Cdk2 and control its activity. Ccna1 is chiefly expressed in meiotic and cancer cell cycles, while A2 and E1 drive the G1 to S phase transition and the S phase events of centriole and DNA duplication (*Heim et al., 2017*). We examined the expression of *Ccna1*, *Ccna2*, *Ccne1* and *Ccne2* during motile ciliogenesis (*Figure 5A*) and detected a large increase, followed by a decline in expression for *Ccna1* and a more modest increase in expression for *Ccne1*. The timing of their peak expression correlates with the indicators of Cdk2 activation during ciliogenesis (*Figure 4A–C*). *Ccne2* did not show statistically significant change and consistent with a postmitotic state, only a negligible amount of *Ccna2* was detected (*Figure 5A*). Our observations confirm

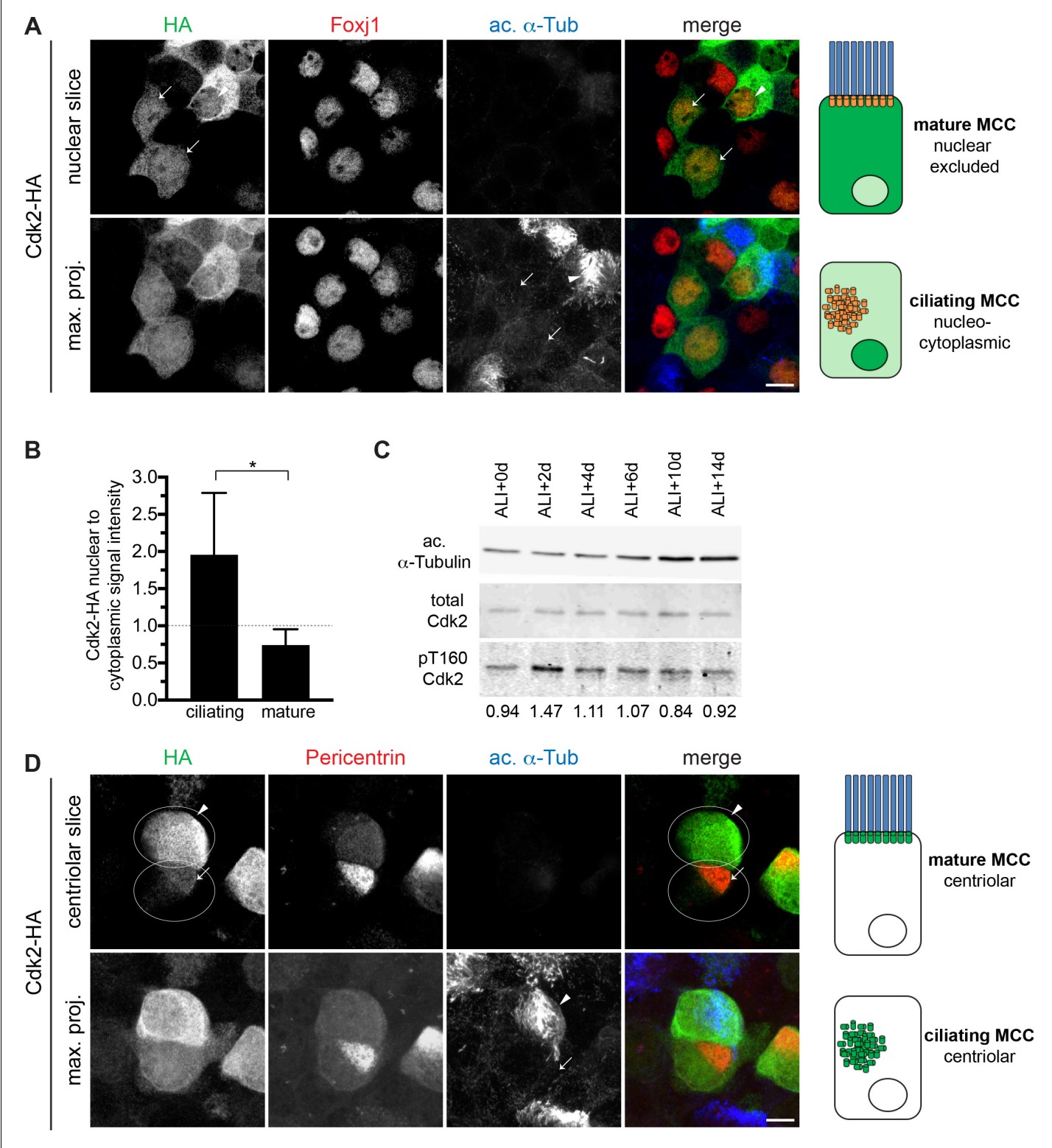

**Figure 4.** Cdk2 is nuclear and active during motile ciliogenesis. (**A**) MTECs were infected with lentivirus encoding HA-tagged Cdk2 at ALI-2d, then labeled at ALI + 4 d with HA (green), Foxj1 (red) and ac. α-Tub (blue) antibodies. A single image slice through the nuclear region (top panel) shows that Cdk2-HA is present in the nucleus in ciliating MCCs (Foxj1+, ac. α-Tub-, arrow) and nuclear excluded in mature MCCs (Foxj1+, ac. α-Tub+, arrowhead). Ac. α-Tub signal visible on the maximum projection (bottom panel) shows cilia on the mature (arrowhead) or lack thereof on the ciliating cells (arrow). Scale bar, 10 μm. (**B**) Quantitation of nuclear to cytoplasmic ratio of Cdk2-HA signal intensity in MCCs indicating nuclear enrichment in ciliating cells.
*Figure 4 continued on next page*

*Figure 4 continued*

Ciliating vs. mature MCCs were identified based on ac. α-Tub and Foxj1 signal. *p<0.0005 C. Western blot of MTEC timecourse lysates shows that total Cdk2 levels are equally abundant at all times, but phospho-T160 Cdk2 is enriched during early ciliogenesis; ratio of phospho/total Cdk2 is indicated under each lane. α-Tubulin signal reflects increasing ciliogenesis and is not to be interpreted as loading control; see *Figure 4—figure supplement 1* for Ponceau S stain for blot which serves loading control. (C) MTECs were infected with lentivirus encoding HA-tagged Cdk2 at ALI-2d, then labeled at ALI + 4 d with HA (green), Pericentrin (red) and ac. α-Tub (blue) antibodies. Image slices through the basal body/centriolar region (top panel) shows that Cdk2-HA is centriolar in both ciliating (arrow) and mature (arrowhead) MCCs. Pericentrin signal is strong on ciliating centrioles and weak on mature basal bodies. Ac. α-Tub signal indicating the mature MCC (arrowhead) is visible on the maximum projection (bottom panel). Scale bar, 5 μm.
DOI: https://doi.org/10.7554/eLife.36375.008

The following source data and figure supplement are available for figure 4:

**Source data 1.** Quantitation of Cdk2-HA nucleo-cytoplasmic localization in ciliating and mature MCCs.
DOI: https://doi.org/10.7554/eLife.36375.010

**Figure supplement 1.** *Cdk1* and *Cdk2* gene expression levels do not change during motile ciliogenesis.
DOI: https://doi.org/10.7554/eLife.36375.009

results from previously published microarray data sets that also identified high *Ccna1* expression during ciliogenesis (*Stubbs et al., 2008*; *Hoh et al., 2012*). The majority of ciliary genes show strong upregulation followed by a decline in expression during ciliogenesis (ex. *Cetn2*), and this pattern can be used to identify transcripts important for the process (*Hoh et al., 2012*). Although we cannot rule out a role for the other canonical cyclins, only *Ccna1* matched this profile.

In addition to *Ccna1* transcripts, we also found an enrichment of the Ccna1 protein during ciliogenesis (*Figure 5B* and *Figure 5—figure supplement 1B*). To test if *Ccna1* expression is restricted to MCCs, we carried out qRT-PCR in sorted MCCs and non-MCCs obtained by FACS from the *Foxj1-EGFP* mouse line (*Ostrowski et al., 2003*) and found that *Ccna1* expression was restricted to MCCs (*Figure 5C* and *Figure 5—figure supplement 1A*). This analysis also revealed that the negligible amount of *Ccna2* expression derives from non-MCCs and that *Ccne1* was detected in both populations (*Figure 5—figure supplement 1A*). We hypothesize that a small population of proliferating cells, likely progenitor basal cells underlying the luminal MCCs or possibly contaminating fibroblasts sometimes observed in the basal regions of MTEC cultures may be the source of these non-MCC transcripts. The MCC-specific pool of *Ccne1* also showed the characteristic rise and fall of expression during ciliogenesis (*Figure 5—figure supplement 1A*), suggesting that *Ccne1* may also regulate ciliogenesis.

Consistent with its MCC-specific expression, we demonstrated that a genomic fragment within the human *CCNA1* promoter can drive Luciferase reporter gene expression in response to the MCC TFs E2F4 (but only in the presence of MCIDAS), FOXJ1 and MYB (*Figure 5D* and *Figure 5—figure supplement 1C*) in 293T/17 cells, similar to a *FOXJ1* promoter fragment previously shown to display MCC-restricted expression (*Tan et al., 2013*). Furthermore, transfection of the same MCC TFs into 293T/17 cells turned on the expression of endogenous human *CCNA1*, and also *FOXJ1* and *CETN2* (*Figure 5E*). *Ccna1* is therefore a target of the MCC gene expression program during ciliogenesis. We hypothesize that employing this A-type cyclin to drive a noncanonical somatic event may drive centriole assembly in MCCs, but not other Cdk2-driven events such as DNA replication.

## Ccna1 localization depends on Cdk2

To investigate whether Ccna1 acts together with Cdk2, we examined the localization of endogenous and lentivirally expressed Ccna1-GFP. We found that similar to Cdk2, Ccna1-GFP was present in the nucleus during ciliogenesis. It was strongly nuclear in ciliating MCCs (*Figure 6A*) while mature MCCs had much lower overall levels of Ccna1-GFP in both nucleus and cytoplasm. Ccna2-GFP was localized in different compartments in different cells and the E-type cyclins showed only nuclear localization at all stages (*Figure 6—figure supplement 1A*). Using a Ccna1 antibody that specifically recognized Ccna1-GFP by Western blot and immunolabeling (*Figure 6—figure supplement 1B–C*), we confirmed the nuclear localization of endogenous Ccna1 in ciliating MCCs. Unlike with Ccna1-GFP, we did not detect endogenous signal in mature MCCs (*Figure 6B*), which may be due to antibody issues. Alternatively, it may indicate the proteasomal degradation of Ccna1 at the end of ciliogenesis, which may be overcome to some extent by continuous lentiviral expression.

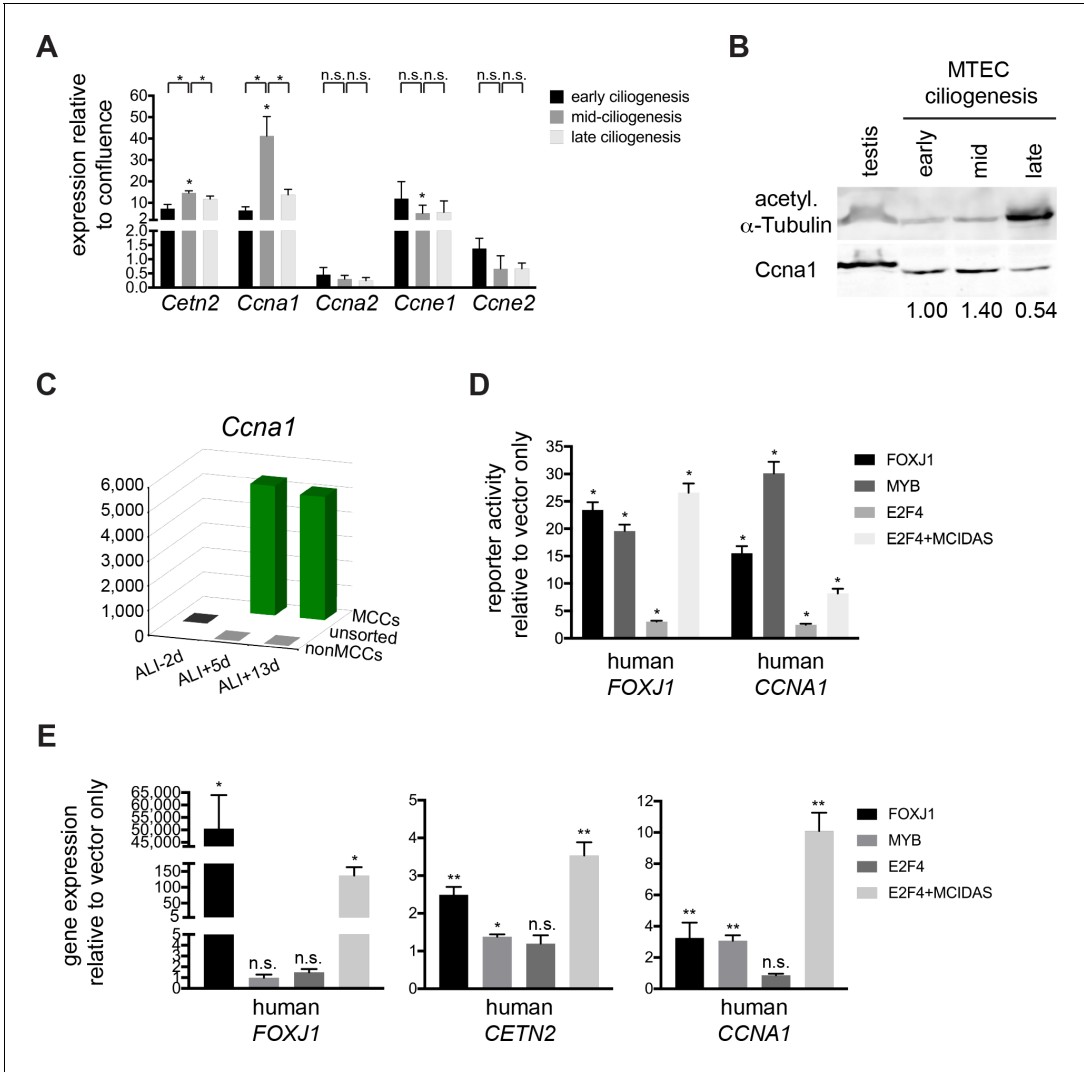

**Figure 5.** *Ccna1* is enriched in MCCs and is a target of the MCC gene expression program during ciliogenesis. (**A**) Quantitative realtime PCR was used to assess A and E-type *cyclin* gene expression during motile ciliogenesis. Similar to the ciliary component *Cetn2*, *Ccna1* and *Ccne1* are enriched during ciliogenesis. *Ccna2* and *Ccne2* are not enriched. Levels were normalized to *Gapdh* expression and compared to values obtained from confluent, ALI-1d samples. Brackets indicate comparison between ciliogenesis timepoints; asterisk above bar indicates significant increase in expression during mid-ciliogenesis compared to confluent, ALI-1d sample. n.s., not significant; *p<0.05 (**B**) Western blot with MTEC lysates using Ccna1 antibody indicates that Ccna1 is enriched in ciliating MTECs. Testis lysate serves as control of Ccna1 expression. Values normalized to early ciliogenesis indicated under each lane. Acetylated α-Tubulin signal reflects increasing ciliogenesis and should not be interpreted as loading control; see *Figure 5—figure supplement 1* for Ponceau S stain for blot, which serves loading control. (**C**) Realtime PCR for *Ccna1* expression in MCCs vs. non-MCCs (sorted from *Foxj1-EGFP* MTECs) shows that it is restricted to MCCs and *Ccna1* expression is higher mid-ciliogenesis (ALI +5 d). (**D**) Luciferase reporter assay using the human *FOXJ1* (left) or *CCNA1* promoter (right) in 293T/17 cells shows that they are responsive to MCC transcriptional regulators compared to vector only control. Promoters are only responsive to E2F4 in the presence of the Mcidas transcriptional activator. *p<0.01 (**E**) 293T/17 cells were infected with lentivirus expressing MCC TFs. Realtime PCR indicates that at least one or more MCC TFs can activate endogenous *FOXJ1* (left), *CETN2* (center) and *CCNA1* (right) gene expression. E2F4 can only activate MCC-specific gene expression in the presence of the Mcidas transcriptional activator. n.s., not significant; *p<0.05, **p<0.01. .

DOI: https://doi.org/10.7554/eLife.36375.011

The following figure supplement is available for figure 5:

**Figure supplement 1.** *Ccna1* expression is restricted to MCCs and is a target of the MCC gene expression program.

DOI: https://doi.org/10.7554/eLife.36375.012

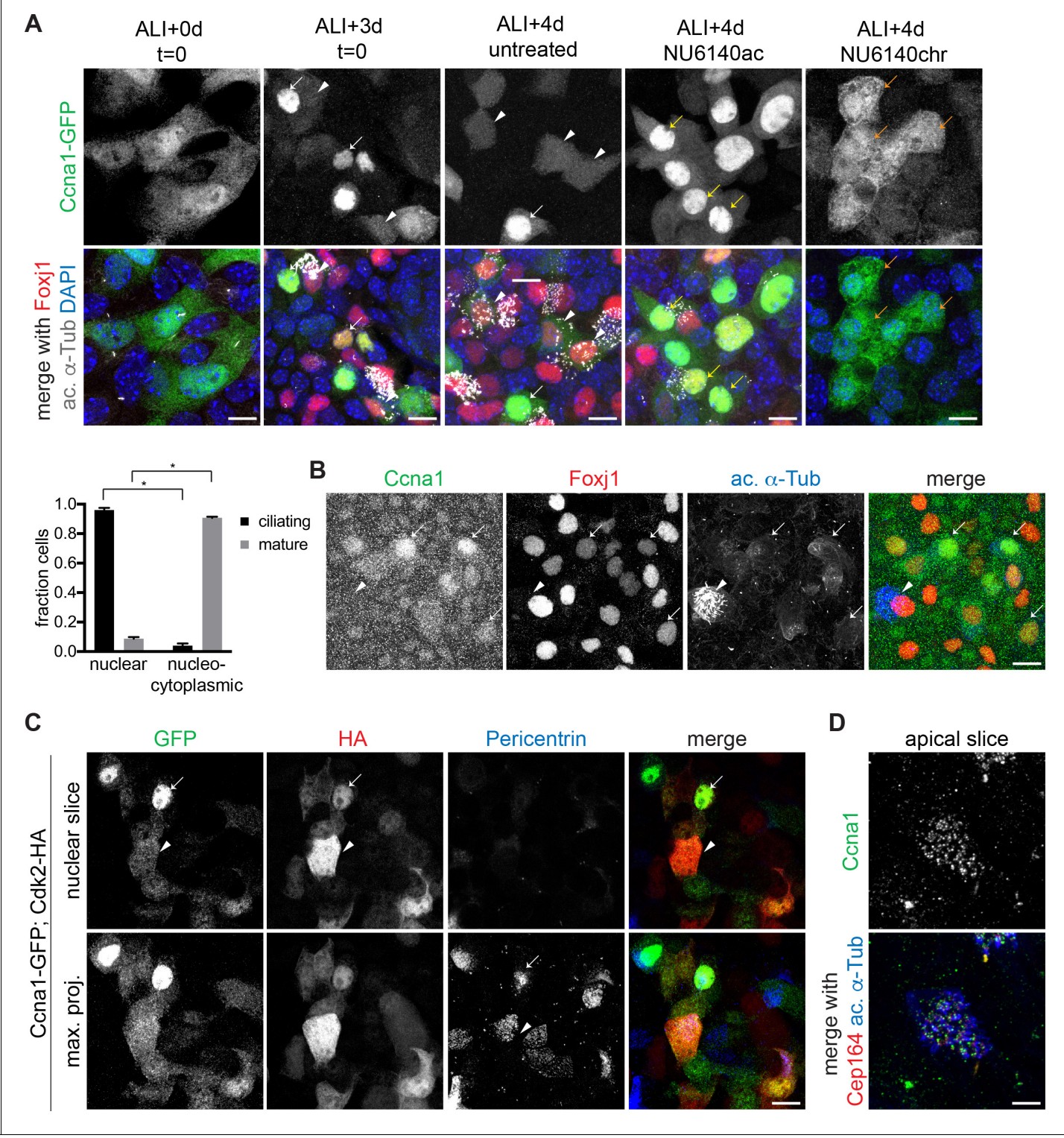

**Figure 6.** Ccna1 is nuclear and localization depends on Cdk activity during ciliogenesis. (**A**) MTECs were infected with lentivirus encoding GFP-tagged Ccna1 at ALI-2d, treated with NU6140 from ALI + 3 to+4 d (NU6140ac, acute treatment) or from ALI + 0 to+4 d (NU6140chr, chronic treatment) then labeled at ALI + 4 d with DAPI (blue), GFP (green), Foxj1 (red) and ac. α-Tub (white) antibodies. The ALI + 3 and 4d untreated panels shows that Ccna1-GFP is nuclear in ciliating MCCs (white arrow) and present in low amounts everywhere in mature MCCs (white arrowhead). Ccna1-GFP is retained in the nucleus in cells arrested during ciliogenesis due to acute NU6140 treatment (yellow arrow). Ccna1-GFP remains nucleo-cytoplasmic in cells blocked from ciliogenesis due to chronic NU6140 treatment (orange arrow). MCC fraction was quantitated based on ac. α-Tub antibody labeling. Scale bar, 10

*Figure 6 continued on next page*

*Figure 6 continued*

µm. *p<0.001 (**B**) ALI + 4 d MTECs labeled with Ccna1 (green), Foxj1 (red) and ac. α-Tub (blue) antibodies shows nuclear Ccna1 (arrow) in ciliating MCCs and no discernible specific signal in mature MCCs (arrowhead). Scale bar, 10 µm. (**C**) MTECs were infected with equal amounts of lentivirus encoding Ccna1-GFP and Cdk2-HA at ALI-2d, then labeled at ALI + 4 d with GFP (green), HA (red) and Pericentrin (blue) antibodies. A single image slice through the nuclear region (top panel) shows that Ccna1-GFP and Cdk2-HA are nuclear in ciliating MCCs (arrow) and nucleo-cytoplasmic in mature MCCs (arrowhead). Centriolar Pericentrin signal shows a tight cluster of centrioles (arrow) in a ciliating MCC and centrioles distributed at the apical surface in a mature MCC. Scale bar, 10 µm. (**D**) ALI + 4 d MTECs labeled at ALI + 4 d with Ccna1 (green), Cep164 (red) and ac. α-Tub (blue) antibodies shows a mature MCC with centriolar Ccna1 signal. Scale bar, 5 µm. .
DOI: https://doi.org/10.7554/eLife.36375.013

The following figure supplement is available for figure 6:

**Figure supplement 1.** Cyclin localization and expression during motile ciliogenesis and Ccna1 antibody validation.
DOI: https://doi.org/10.7554/eLife.36375.014

We found that similar to other ciliary genes, endogenous *Ccna1* gene expression is suppressed by both chronic (ALI + 0 to+4 d) and acute (ALI + 3 to+4 d) Cdki treatment (*Figure 6—figure supplement 1D*). Upon lentiviral co-expression, Cdk2-HA and Ccna1-GFP colocalized in the nucleus and Ccna1 was also detected at the centrioles (*Figure 6C–D*). Moreover, we found that Ccna1-GFP localization depends on Cdk2 activity (*Figure 6A*), as Ccna1-GFP failed to accumulate in the nucleus in MTECs under chronic Cdki treatment that blocks the initiation of motile ciliogenesis, nor was it able to exit the nucleus under acute Cdk inhibition that arrests MCCs at intermediate stages of ciliogenesis. These results support a model in which, as in cycling cells, Cdk2 and Ccna1 function as a complex to regulate motile ciliogenesis in MCCs.

## Ccna1 mutant mice have fewer MCCs

*Ccna1* knockout mice are viable, but male-sterile due to the requirement for Ccna1 in the meiotic divisions of the male germline (*Liu et al., 1998*). To assess a potential requirement for Ccna1 in the regulation of motile ciliogenesis, we observed MCCs by scanning electron microscopy and ac. α-Tub antibody labeling in the trachea and bronchi of adult *Ccna1$^{-/-}$* mice. We consistently observed a 2.46-fold reduction in the fraction of MCCs in the mutant airways compared to wildtype litter mates (0.15 ± 0.01 vs. 0.36 ± 0.01). *Ccna1$^{-/-}$* epithelia also contained an increased number of dome-shaped, possibly secretory cells, which may indicate further dysfunction in these airways (*Figure 7A*, *Figure 7—figure supplement 1A*). In comparison, we examined *Ccne1; Ccne2* double knockout mice (*Geng et al., 2003*) and found no difference in the fraction of MCCs. (*Figure 7B*). Based on these observations, we conclude that Ccna1 is required for motile ciliogenesis. Although the ectopic expression of Ccna1 and Cdk2 either separately or together is not sufficient to drive the motile ciliogenesis pathway (*Figure 1E*, *Figure 7—figure supplement 1B* and not shown), our results are consistent with Cdk2 acting together with Ccna1 and possibly other cyclins upon MCC fate acquisition to activate the MCC gene expression program.

## Discussion

Previously, the earliest known event in the motile ciliogenesis pathway was the transcriptional upregulation of ciliary genes by MCC TFs that are known to also regulate the cell cycle in dividing cells. Here, we show that Cdk2, the key regulator of cell cycle entry and centrosome and DNA duplication in S phase is responsible for activation of the MCC gene expression program (*Figure 7C*), further supporting the notion that an alternative cell cycle program controls MCC differentiation. Seeking to understand how differentiated MCCs can overcome the strict regulatory limits on centriole duplication that exist in cycling cells to generate hundreds of centrioles, we tested the requirement for Cdk2 during motile ciliogenesis. We found that it is required for initiating and sustaining the MCC gene expression program, and that it likely works together with Ccna1. These results uncover surprising roles for both Cdk2 and Ccna1 in a quiescent somatic cell, further establish the role of cell cycle regulators in motile ciliogenesis, and suggest that centriole generation and number control are regulated by common cell cycle-associated mechanisms in which MCC-specific alterations can drive amplification.

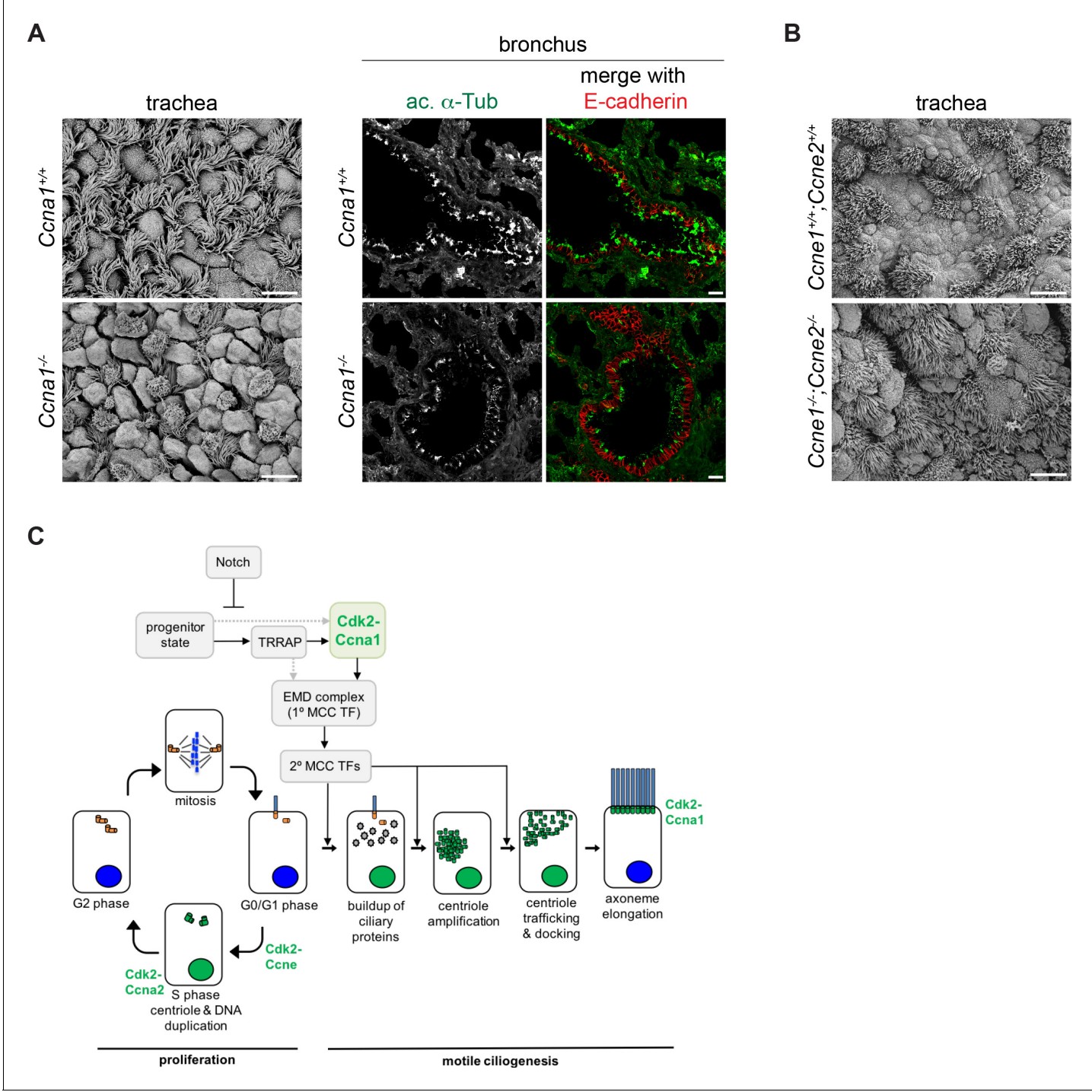

**Figure 7.** *Ccna1*, but not *Ccne1;Ccne2* mutant mice have fewer MCCs. (**A**) SEM of adult *Ccna1⁻/⁻* trachea (left panels) and ac. α-Tub (green) antibody labeling of bronchi (epithelium marked with E-cadherin antibody labeling, red) on cryosectioned lung tissue (right panels) show fewer MCCs with shorter, sparser cilia (bottom) compared to wildtype littermates (top). See *Figure 7—figure supplement 1A* for quantitation of MCC fraction for SEM. Scale bar, 10 μm for trachea, 20 μm for lung. (**B**) SEM of *Ccne1⁻/⁻;Ccne2⁻/⁻* trachea shows no difference in the number and distribution of MCCs (bottom) compared to wildtype littermates (top). Scale bar, 10 μm. (**C**) Schematic of Cdk2 function during cell cycle progression and proposed Cdk2-Ccna1 activity in the motile ciliogenesis pathway. Solid and dotted arrows linking the progenitor state to EMD activation represent alternative potential pathways that have yet to be distinguished. .

DOI: https://doi.org/10.7554/eLife.36375.015

The following source data and figure supplement are available for figure 7:

*Figure 7 continued on next page*

*Figure 7 continued*

**Source data 1.** Quantitation of MCC fraction in Ccna1 mutant and control mice.
DOI: https://doi.org/10.7554/eLife.36375.017
**Figure supplement 1.** Ectopic Ccna1 expression is not sufficient to drive motile ciliogenesis.
DOI: https://doi.org/10.7554/eLife.36375.016

Centriole generation is normally restricted to the S phase of the cell cycle. The involvement of multiple cell cycle-related proteins in the motile ciliogenesis pathway led to the hypothesis that MCCs enter a unique cell cycle state, the so called S* phase (*Tan et al., 2013*) that shares characteristics with both cycling and quiescent states and simultaneously facilitates both centriole generation and maintenance of a postmitotic state. Thus, we speculated that Cdk2, the regulator of S phase entry may be involved in ciliogenesis. We report that Cdk2 is localized to the nucleus in ciliating MCCs, and that Cdk2 inhibition arrests the pathway at its earliest known step, the initiation of the MCC gene expression program. Nuclear localization is consistent with an active kinase (*Pagano et al., 1993*) and a role in regulating MCC transcription.

MCC TFs represent attractive candidate substrates for Cdk2 during ciliogenesis. Such a direct action mechanism would be consistent with both the lack of transcriptional activation under Cdki treatment and with known functions of Cdk2 in dividing cells. Cdk2-Ccne1 propels cells towards S phase, in part, by phosphorylating the Rb protein, which maintains the E2F1 transcription factor in an inactive complex with Dp1 (*Heim et al., 2017*). Related complexes, comprising E2F4/5, Dp1 and a Geminin family member (Geminin, Mcidas and Gmnc) act early during ciliogenesis to turn on MCC gene expression (*Vladar and Mitchell, 2016*). Recent studies on the role of the Geminin family proteins in MCCs suggest that Geminin is initially complexed with E2F4/5 and Dp1 to inhibit ciliogenesis; subsequently, complexes that include Mcidas and Gmnc emerge to turn on ciliary genes. We favor the hypothesis that this process is regulated by Cdk2 phosphorylation of one or more of these complex components. E2F5 is a known Cdk2 target (*Morris et al., 2000*), and studies are underway to test whether this interaction occurs during ciliogenesis.

Our results raise a number of fascinating questions about Cdk2 specifically and about cell cycle regulation in general. How does Cdk2 become active in a quiescent cell? How does Cdk2 activation in MCCs not lead to cell cycle reentry or to DNA replication? We hypothesize that Cdk2 activation during ciliogenesis occurs in the proposed S* phase that shares characteristics with both G1 and S. Moreover, we propose that events downstream of Cdk2 activation are modular and subject to MCC-specific regulation to allow centriole amplification without DNA replication or cell cycle progression. As the key regulator of the G1 to S phase transition in cycling cells, Cdk2 activity is under a multitude of regulatory constraints, and our results indicate that some of these regulatory events are also active in MCCs. We demonstrated that Cdk2 bearing the activating Thr160 phosphorylation peaks around the time of ciliogenesis initiation in MTECs. In dividing cells, this modification is imparted by Cdk-activating kinase (CAK) downstream of cyclin binding (*Jeffrey et al., 1995*), and it is essential for the G1 to S phase transition. In the future, we will test the potential activity and role of additional Cdk2 regulatory measures, including the removal of the inhibitory Thr14 and Tyr15 phosphorylations by the Cdc25A phosphatase (*Heim et al., 2017*).

Our data point to the involvement of Ccna1 as a cognate cyclin to Cdk2 in motile ciliogenesis regulation based on its MCC restricted expression, colocalization with Cdk2 and the finding of diminished MCCs in the *Ccna1* mutant mice. Although technical limitations (the inability to sort ciliating cells from heterogeneous cultures, and the small culture size) prevent us from demonstrating direct association in ciliating cells, their colocalization during ciliogenesis, similar loss of function phenotypes and the existence of Cdk2-Ccna1 complexes in other tissues (*Joshi et al., 2009*) are consistent with Ccna1 binding to activate Cdk2 to control ciliogenesis. Finally, we show that Cdk2 and Ccna1 are both enriched in nuclei of ciliating but not mature MCCs. Of note, Ccna1 regulates meiosis in mammals and plants (*Liu et al., 1998*; *d'Erfurth et al., 2010*), another event in which centrioles are generated in the absence of DNA replication.

We demonstrate that *Ccna1* is a target of multiple MCC transcription factors, which accounts for its sustained and robust expression during ciliogenesis. As we propose that Ccna1 acts together with Cdk2 to initiate the MCC gene expression program, it remains to be understood how *Ccna1* gets turned at the onset of ciliogenesis. By analogy to cell cycle progression, we speculate that

Ccna1 may be degraded at or near the conclusion of motile ciliogenesis, possibly as a regulatory step in the pathway. Our inability to detect Ccna1 by antibody labeling in mature MCCs raises the possibility of a regulated cyclin destruction event, which requires closer examination. *Ccna1* mice showed a reduction in MCC number but not the complete block to ciliogenesis observed with Cdki treatment. The different phenotypes could represent different responses to a chronic loss of Ccna1 in vivo compared to the acute inhibition of Cdk activity in vitro. We also speculate that it may be due to partial redundancy with other cyclins. In spite of its broader expression and the lack of an MCC phenotype, we cannot rule out a role for Ccne1. Recent studies identified Ccno as an important regulator of ciliogenesis downstream of the EMD complex (*Wallmeier et al., 2014*), and although it is not known whether Ccno can activate Cdk2, it could conceivably also act partially redundantly with Ccna1.

We consistently observed Cdk2 at centrioles in both MTECs and in human ALI cultures (unpublished results). Cdks, including Cdk2, have been reported at the centrosome in dividing cells, so this is likely a localization pattern shared with MCCs. Centrosomal Cdk2 targets include Ccp110, which dissociates from the mother centriole distal end upon phosphorylation to allow primary cilium growth (*Chen et al., 2002*). Ccp110 is present and required for motile ciliogenesis (*Song et al., 2014*), and the centriolar pool of Cdk2 may be regulating motile axoneme elongation in MCCs. Nucleophosmin 1 (Npm1), a Cdk2 substrate that shows both nuclear and centriolar localization, has been suggested as a link between Cdk2 activity and centrosome duplication (*Okuda et al., 2000*). Although we can detect Npm1 in MCCs (unpublished results), its role in ciliogenesis remains untested. We note that forced expression of Mcidas in the presence of inhibited Cdk2 can sustain ciliogenesis, suggesting that putative Cdk2 activity at replicating centrioles may not be absolutely required.

Our results, and the analogy between the G1 to S phase transition and motile ciliogenesis, indicate that Cdk2 is a common regulator of the two processes, however, it is important to consider the possibility of redundant factors and mechanisms. The near normal viability of the *Cdk2* knockout mouse (*Berthet et al., 2003*) revealed that other Cdks and Cdk-independent events are capable of driving cell cycle progression, at least in the context of chronic Cdk2 absence. Similarly, we cannot rule out the contribution of other Cdks to motile ciliogenesis. Cdk4 and Cdk6 play important roles in G1 exit and should be investigated in MCCs. Our Cdkis likely do not inhibit Cdk4/6 at the concentrations we employed, but some might conceivably target Cdk5 (*Peyressatre et al., 2015*). Although Cdk5 is not a canonical cell cycle regulator (*Shupp et al., 2017*), we cannot exclude that it has a role in motile ciliogenesis.

The Cdk1-APC/C mitotic oscillator was recently shown to promote intermediate phases of centriole replication in MCCs without stimulating mitosis (*Al Jord et al., 2017*). We demonstrate that Cdk2 plays a role in initiating the MCC gene expression program, the precursor to centriole assembly. We support our placement of Cdk2 at this point in the regulatory hierarchy by demonstrating that it acts downstream of the Notch signaling event and either downstream of, or in parallel with, TRRAP, an early regulator of ciliogenesis, and upstream of the EMD complex that initiates and sustains MCC gene expression. In addition to its nuclear localization, we observe Cdk2 at centrioles, and thus we cannot rule out that, similar to Cdk1, it is also involved more directly in ciliogenesis. Cdk1 and Cdk2 redundancy at centrioles may explain why we did not detect a ciliogenesis phenotype with the Cdk1-specific RO3306 inhibitor as observed by Al Jord *et al*. As that study was chiefly carried out in ependymal MCCs, which make many fewer MCCs, it may also reflect an alternative Cdk requirement.

We set out to increase our understanding about how MCCs are able to break the rules of centriole assembly and number control faithfully observed by dividing cells. In sum, our results show that Cdk2 activity is required to create a permissive environment for centriole generation in either context. Emerging studies point to the involvement of universal centriole assembly factors and events, including common regulators and biogenesis steps, and MCC-specific features, like the use of deuterosomes and transcriptional regulation. We add the universal requirement for Cdk2 activity to this model, and underscore the importance of cell cycle regulation to centriole assembly in any context. It will be interesting to compare MCC mechanisms to those operating in other cells with alternative cell cycles or where DNA and centriole duplication are uncoupled. These include male germ cells in which meiosis is regulated by Cdk2-Ccna1 (*Liu et al., 1998*) and endocycling cells that undergo many rounds of DNA replication without mitosis (*Lu and Roy, 2014*). With growing attention to

MCCs in various tissues, improved understanding of the motile ciliogenesis pathway is an important goal.

## Materials and methods

### Mouse husbandry and MTEC culture

C57BL/6J (IMSR Cat# JAX:000664, RRID:IMSR_JAX:000664) mice were obtained from JAX. *Ccna1* (MGI Cat# 2657243, RRID:MGI:2657243) (*Liu et al., 1998*), *Ccne1; Ccne2* (MGI Cat# 2675493, RRID:MGI:2675493, gift from Peter Sicinski, Harvard Medical School, Cambridge, MA) (*Geng et al., 2003*) and *Foxj1-EGFP* (IMSR Cat# JAX:010827, RRID:IMSR_JAX:010827, gift from Larry Ostrowski, UNC Chapel Hill, Chapel Hill, NC) (*Ostrowski et al., 2003*) mice have been previously described. All procedures involving animals were approved by the Institutional Animal Care and Use Committee of Stanford University School of Medicine in accordance with established guidelines for animal care. MTEC culture and lentiviral infection were carried out as previously described (*Vladar and Brody, 2013*). In short, tracheas were isolated and incubated overnight in Pronase solution to release epithelial cells. Cells were seeded onto 24, 12 or 6 well size Transwell filters (Corning) and cultured submerged in proliferation medium until confluence. The air-liquid interface (ALI) was created by adding differentiation medium to only the bottom compartment of the culture dish. Cells were treated at ALI with 10 μM Roscovitine, 5 μM Purvalanol A, 10 μM NU6140, 10 μM Cdk2 Inhibitor III, 1 μM RO3306 (all from Tocris) and 1 μM DAPT (Abcam) in differentiation medium for various lengths of time. MTECs are fed fresh medium with or without drugs every two days.

Lentiviral vectors containing myc-Mcidas, Foxj1 and Myb have been previously described (*Tan et al., 2013*). Lentiviral vectors containing human Cdk2-HA and Cdk2$^{D145N}$-HA (*van den Heuvel and Harlow, 1993*) were generated by transferring the Cdk2-HA open reading frames into the BamHI site of the pRRL.sin-18.PPT.PGK.pre lentiviral vector using PCR. The GFP-Ccna1/a2/e1 and e2 constructs were generated by PCR amplifying and inserting the cyclin open reading frames, obtained by RT-PCR from MTEC or NIH/3T3 cell cDNA, into the AgeI or BamHI site of the pRRL.sin-18.PPT.PGK.GFP.pre lentiviral vector to create a C-terminal GFP fusion. Lentivirus was prepared according to published methods using the psPAX2 and pMD2.G helper plasmids (Addgene) in the 293T/17 cell line (see below). MTECs were infected with lentivirus on day three of culture (ALI-2d) using spin infection following EGTA treatment to temporarily disrupt epithelial junctions (*Vladar and Brody, 2013*).

### Cell lines

293T/17 (ATCC Cat# CRL-11268, RRID:CVCL_1926) cells were used for lentiviral preparation and the Luciferase and endogenous gene activation assays. mIMCD3 (ATCC Cat# CRL-2123, RRID:CVCL_0429) cells were used to test Ccna1 antibody specificity. Cells were purchased from ATCC and were assumed to be authenticated by the supplier. Cells were not specifically monitored for mycoplasma contamination, but routine DAPI staining would have revealed the presence of contamination.

### Immunofluorescence and immunohistochemistry

MTECs grown on 24 well size Transwells were fixed in −20°C methanol or 4% paraformaldehyde for 10 min, blocked in 10% normal horse serum and 0.1% Triton X-100 in PBS and incubated with primary antibodies for 1–2 hr, then with secondary antibodies for 30 min at room temperature. Samples were mounted in Mowiol mounting medium containing 2% N-propyl gallate (Sigma). Lung tissues were fixed in 4% paraformaldehyde overnight at 4°C, rinsed in PBS, incubated in 30% sucrose, embedded in OCT compound and frozen. 10 μm cryosections were labeled as above. Samples were imaged with Leica LAS X software on a Leica SP5 or SP8 confocal microscope (Leica). For antibodies and fixation conditions, see Supplementary Table 1 in *Supplementary file 1*.

### Cell lysates and western blots

MTEC lysates were generated in triplicate from MTECs cultured in 12- or 6-well Transwells to the desired stage and harvested by scraping the Transwell surface in 1x Laemmli sample buffer. Approximately 10,000 cells per lane were loaded for SDS-PAGE, then transferred to nitrocellulose membrane. For the MTEC timecourses equal loading was verified by Ponceau S staining of the

membrane. Acetylated α-Tubulin antibody labeling shows increasing signal with increased ciliogenesis as MTECs mature and thus should not be interpreted as a loading control. Mouse testis lysate was prepared using published methods (*Panigrahi et al., 2012*) to serve as a control for Ccna1 expression.

## Quantitative realtime PCR

cDNA was prepared from MTECs cultured on 12- or 6-well Transwells at various timepoints and with various treatments as indicated, and from MCCs (EGFP+) and non-MCCs (EGFP-) obtained by FACS from *Foxj1-EGFP* MTECs grown on 12 6-well Transwells as previously described (*Vladar and Stearns, 2007*). Early, mid and late ciliogenesis timepoints were obtained at ALI + 2, 4 and 8 days. *Gapdh* levels were used to normalize target gene expression values. Ciliogenesis timecourse gene expression levels were compared to levels in confluent, but not yet differentiated cells harvested at ALI-1d. MCC vs. non-MCC timecourse gene expression levels were compared to levels at ALI + 0 d. qPCR was performed in triplicate with Power SYBR Green Master Mix (Thermo Fisher) in a StepOne-Plus Real-Time PCR System (Thermo Fisher), and gene expression was evaluated using the ΔΔCt method. For primer sequences, see Supplementary Table 2 in *Supplementary file 1*.

## Cdk2 nucleo-cytoplasmic signal quantitation

The ratio of lentivirally expressed Cdk2-HA nuclear to cytoplasmic signal intensity in MCCs (identified by Foxj1 and ac. α-Tub signal) was quantitated on individual image slices using ImageJ (NIH) by measuring signal intensity for manually outlined nuclear and cytoplasmic areas (nuclear area identified by Foxj1 or DAPI signal) and normalizing to the total measured area.

## Luciferase assay

Human FOXJ1 and CCNA1 promoter genomic DNA fragments (see *Figure 5—figure supplement 1B*) were cloned into the pGL4.20 (Promega) firefly luciferase expression vector and transfected into 293T/17 cells using FuGENE6 (Roche) along with plasmids containing human FOXJ1, MYB, E2F4 and/or MCIDAS cDNAs (*Tan et al., 2013*) and pRL-TK (Promega) Renilla luciferase expression vector. Reporter activity was assessed using the Dual-Glo Luciferase Assay System (Promega) with a FLUOStar Omega (BMG Labtech) luminescence plate reader. Relative reporter activity was calculated by normalization to the vector only transfection control in triplicate.

## Electron microscopy

Adult mouse airway tissues were fixed in 2% glutaraldehyde, 4% paraformaldehyde in 0.1M Sodium Cacodylate buffer, pH 7.4 (all from Electron Microscopy Sciences) at 4°C overnight, osmicated, dehydrated and dried with a Tousimis Autosamdri-815 critical point dryer. Samples were then mounted luminal side up, sputter coated with 100 Å layer of Au/Pd and analyzed with a Hitachi S-3400N VP-SEM microscope (Hitachi) operated at 10–15 kV, with a working distance of 7–10 mm and using secondary electron detection.

## Transparent reporting

1. Sample-size estimation: No explicit power analysis was used during the design of the study. Biological and experimental replicates (indicated in Supplementary Table 3 in *Supplementary file 1*) were completed to at least an n = 3 where possible.
2. Replicates: replicate information is found in Supplementary Table 3 in *Supplementary file 1* with biological replicates indicated by underlining and technical replicates indicated by italicized font. Outliers were not encountered. Data were not excluded from analysis.
3. Statistical reporting: the Prism7 software (GraphPad Software) was used to generate graphs and perform statistical analyses. Pairwise comparisons were made with a two-tailed Student's t test or an ANOVA test. For multiple comparisons, a follow up test (Dunnett's or Bonferroni's) was applied to correct for multiple hypothesis testing. For all cases, a p value less than 0.05 was considered significant. Error bars on graphs represent standard error. See Supplementary Table 3 in *Supplementary file 1* for information about replicates and statistical tests for data presented in figures.
4. Group allocation: samples generated in an identical manner were allocated into groups based on treatment (drug, lentiviral gene expression, etc.) or genotype.

5. Source data: not applicable

## Acknowledgements

We thank Klara Fekete for help with animal husbandry, Koshi Kunimoto for help with histology, Lydia Joubert (Stanford Cell Sciences Imaging Facility) for help with electron microscopy, Aron Jaffe (Novartis) for the TRRAP antibody, and Chris Kintner (Salk) and members of the Axelrod and Stearns labs for helpful discussions. Work was supported by R01GM098582 (NIH) to JDA, R01GM052022 and R01GM121424 (NIH) to TPS and 1R01HD034915 (NIH) to DJW. EKV is a Boettcher Foundation Webb-Waring Early Career Investigator.

## Additional information

### Funding

| Funder | Grant reference number | Author |
|---|---|---|
| National Institutes of Health | R01GM052022 | Tim Stearns |
| National Institutes of Health | R01GM121424 | Tim Stearns |
| National Institutes of Health | R01GM098582 | Jeffrey D Axelrod |
| National Institutes of Health | 1R01HD034915 | Debra Wolgemuth |

The funders had no role in study design, data collection and interpretation, or the decision to submit the work for publication.

### Author contributions

Eszter K Vladar, Conceptualization, Data curation, Formal analysis, Investigation, Methodology, Writing—original draft, Writing—review and editing; Miranda B Stratton, Maxwell L Saal, Glicella Salazar-De Simone, Xiangyuan Wang, Investigation, Writing—review and editing; Debra Wolgemuth, Funding acquisition, Writing—review and editing; Tim Stearns, Jeffrey D Axelrod, Supervision, Funding acquisition, Writing—review and editing

### Author ORCIDs

Eszter K Vladar http://orcid.org/0000-0002-4160-8894
Tim Stearns http://orcid.org/0000-0002-0671-6582

### Ethics

Animal experimentation: All procedures involving animals were approved by the Institutional Animal Care and Use Committee of Stanford University School of Medicine (#17926) in accordance with established guidelines for animal care.

### Decision letter and Author response

Decision letter https://doi.org/10.7554/eLife.36375.021
Author response https://doi.org/10.7554/eLife.36375.022

## Additional files

### Supplementary files

• Supplementary file 1. Supplementary Table 1 Antibodies used. Supplementary Table 2 Quantitative realtime PCR primers used. All primers amplify the mouse target unless otherwise specified. Supplementary Table 3 Statistical test and replicate information for data.
DOI: https://doi.org/10.7554/eLife.36375.018

• Transparent reporting form
DOI: https://doi.org/10.7554/eLife.36375.019

## Data availability

All data generated or analysed during this study are included in the manuscript and supporting files.

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
