## [Decision Letter]

Thank you for submitting your article "Cyclin-dependent kinase control of motile ciliogenesis" for consideration by *eLife*. Your article has been reviewed by three peer reviewers, one of whom is a member of our Board of Reviewing Editors, and the evaluation has been overseen by Anna Akhmanova as the Senior Editor. The following individual involved in review of your submission has agreed to reveal his identity: Steven Brody (Reviewer #3).

The reviewers have discussed the reviews with one another and the Reviewing Editor has drafted this decision to help you prepare a revised submission.

Summary

The manuscript submitted by Vladar and colleagues reports a novel role for the canonical mitotic regulator Cdk2 in the control of ciliogenesis in murine airway culture systems. They claim that Cdk2 is required to initiate and maintain the MCC gene expression program during ciliogenesis. They claim that Cdk2 is coupled with cyclin A1 to initiate centriole replication in MCCs. The authors begin by assessing the effects of pharmacologic Cdk inhibitors on differentiating MTEC ALI cultures, observing that inhibition of Cdk2 produces a reversible block in the initiation ciliogenesis revealed by the absence of MCC markers by immunofluorescence imaging. The specificity of this result was established with a dominant-negative allele of Cdk2, which phenocopies pharmacologic inhibition. Subsequent data suggests ongoing Cdk2 activity (after initiation of ciliogenesis) is also required for MCCs to complete maturation into "Stage IV" or Foxj1+/Myb- cells. In order to establish epistasis relationships, MTEC ALI were treated simultaneously with Cdk2 inhibitors and the Notch antagonist DAPT or with Cdk2 inhibitors in the setting of enforced Mcidas expression. The claim is that these experiments place Cdk2 downstream of Notch and upstream of Mcidas in specifying MCCs. In order to identify the cellular compartment in which Cdk2 functions, tagged overexpressed Cdk2 was localized at ALI+4d, a timepoint at which both mature and ciliating MCCs are present. The authors claim nuclear exclusion of HA-Cdk2 in mature (but not ciliating) MCCs and go on to co-localize HA-Cdk2 to the centriolar compartment. In order to identify the cognate cyclin with which Cdk2 associates during MCC development, the authors utilize qRT-PCR and note the potent induction of Ccna1 during the mid-ciliogenesis stage specifically in MCCs (defined here by Foxj1 expression). Reporter assays and endogenous qRT-PCR experiments upon enforced overexpression of MCC transcription factors suggest Ccna1 can be induced by E2F4 in the presence of Mcidas. The authors next examine the localization of ectopic and native Ccna1, concluding Ccna1 is nuclear localized during ciliogenesis and subsequently undetectable or nucleocytoplasmic in mature MCCs. Given the similarity of this pattern to that previously observed for Cdk2, they argue for the function of Cdk2/Ccna1 as a physically interacting complex. In the final series of experiments the authors demonstrate the in vivo importance of Ccna1 in ciliogenesis by examining the airways of previously generated *Ccna1^-/-^* mice.

All the reviewers agree that there is broad general interest in these findings. It is clear that Cdk2/Ccna1 are important regulators of airway ciliogenesis. The data are convincing that Cdk inhibitors interfere with both processes: centriologenesis and cilia formation. This is an important contribution. However, there are substantive concerns that need to be addressed about the interpretation and mechanism and the various steps of ciliogenesis and maintenance that are affected by Cdk2.

Major Concerns that must be addressed:

1) The actual role of Cdk2 in ciliated cell formation, ciliogenesis, and cilia maintenance per se is unclear. They need to clarify effects on cell fate specification and ciliogenesis and ciliated cell maintenance. Is Cdk2 directing specifically the initiation and (perhaps) maintenance of MCC centriologenesis or is there a broader role in the (co-)regulation of other steps of cilia assembly. A specific role in centriole replication needs to be clear. It may be that there are biological or technical limitation such that a failure of initiation of centriole amplification halts the entire subsequent program of ciliogenesis, this may be difficult to prove in a revision. However, the apparent later stage role of Ccna1, after centriologenesis, seems contradictory to the proposal that Cdk2-Ccna1 are regulating centriole replication and needs to be resolved. There might be some experimental limitations that cannot fully separate the roles of Cdk2 in regulating ciliated cell formation and centriologenesis. But the authors need to clarify the specific roles of Cdk in either dilated cell formation or centriologenesis and also discuss these limitations in the paper

2) The links between the celluar localization of Cdk2, the activation of Cdk2, and the expression of MCC TFs are inconclusive. It is difficult to appreciate the nuclear-cytoplasmic shuttling during ciliogenesis. In Figure 4, nuclear vs cytoplasmic lysates would be better than this IF (which isn't convincing for nuclear exclusion). One could immunoprecipitate tagged Cdk2 from nuclear and cytoplasmic compartments at various timepoints and show Ccna1 blots?

3) The experimental evidence is not sufficient to support that CdK2 acts downstream of Notch signaling. The authors cannot exclude the possibility that Cdk2 and Notch signaling function parallel but share the same downstream consequence.

Requirements for revision:

1) The authors should provide additional evidence that Cdk2 acts downstream of Notch signaling, rather than in a parallel pathway. A further assessment of the components of Notch signaling might be one approach.

2) The authors should provide more convincing evidence of nuclear-cytoplasmic shuttling with the HA-Cdk2, in control, basal cells and in cells undergoing centriologenesis and ciliogenesis. Identification of the fusion protein in Isolated nuclear and cytoplasmic fractions could be performed.

3) Biochemical evidence of the interaction of Cdk2 with Ccna2 in cyotoplasmic fractions during specific stages of centriologenesis and/or cilia assembly. Related to this strengthen evidence regarding the relationship of HA-Cdk2 and components of centrioles (biochemically or by IF).

4) A better characterization of pT160 Cdk2 to identify which specific stages of MCC differentiation are associated with this activity. For example, is there coincident biochemical evidence of high levels of the massive centriole replication (and the presence of other markers of centriologenesis) or it later, post centriole replication/docking and instead coincident with cilia assembly? Or, during both processes?

5) There are several relatively minor points that ought to be addressed related to clarifying the use of specific parameters in graphs, providing statistics, controls for western blots, and quantifying ciliated vs non-ciliated cells in Ccna1 ko mice.

6) Finally, the last four paragraphs of the Discussion could be condensed since it is speculation and not directly focused on the data. I think that one major emphasis of the discussion should be a comparison to Al Jord et al., 2017, who show that Cdk1 (rather than Cdk2) in balance with APC is required for centriole amplification-ciliation and inhibition of mitosis, respectively. The Vlader discussion should further explain why Al Jord implicates Cdk1 and Vlader Cdk2 beyond what is written in paragraph two. The authors should discuss specifically if they believe that their data indicate that Cdk2 is directly a program or has specific protein targets, these issues are discussed loosely in paragraphs three, four and five. Consider inhibition of APC and examine Cdk2-dependent centriole amplification, this would address the discussion point in paragraph six.

---

## [Author Response]

Major Concerns that must be addressed:1) The actual role of Cdk2 in ciliated cell formation, ciliogenesis, and cilia maintenance per se is unclear. They need to clarify effects on cell fate specification and ciliogenesis and ciliated cell maintenance. Is Cdk2 directing specifically the initiation and (perhaps) maintenance of MCC centriologenesis or is there a broader role in the (co-)regulation of other steps of cilia assembly. A specific role in centriole replication needs to be clear. It may be that there are biological or technical limitation such that a failure of initiation of centriole amplification halts the entire subsequent program of ciliogenesis, this may be difficult to prove in a revision. However, the apparent later stage role of Ccna1, after centriologenesis, seems contradictory to the proposal that Cdk2-Ccna1 are regulating centriole replication and needs to be resolved. There might be some experimental limitations that cannot fully separate the roles of Cdk2 in regulating ciliated cell formation and centriologenesis. But the authors need to clarify the specific roles of Cdk in either dilated cell formation or centriologenesis and also discuss these limitations in the paper

This is a complex multi-part question, which we attempt to break down and address systematically.

First, it is apparent that we failed to adequately convey the currently accepted understanding of these events. We hope that the following is a clearer description of the events and regulation of cell fate specification, motile ciliogenesis and cilium maintenance during multiciliated cell (MCC) differentiation. To amend the manuscript, we have added two new schematics to Figure 1—figure supplement 1 (new Figure 1—figure supplement 1A-B), updated Figure 1—figure supplement 1C and Figure 7B (now 7C) and clarified the text in the Introduction as well as the Discussion.

To summarize: upon airway epithelialization, MCC and secretory cell fates are acquired in a Notch signaling-dependent mechanism. Cells in which Notch signaling is activated are diverted to the secretory cell fate, whereas cells in which Notch is not activated differentiate into MCCs. Therefore, it is the absence of Notch signaling that leads to the MCC fate. We note that typically, Notch signaling entails a feedback, in which neighboring cells emerge as either Notch expressing cells, in which Notch is activated (here, the secretory cells), or ligand expressing cells, in which Notch is not activated (here, the MCCs). In the prospective MCCs in which Notch is not activated, it appears that MCC differentiation is a default occurrence, though we cannot rule out that some additional initiating event is required that might or might not be linked to having expressed ligand.

Thus, the first key point that we make is that the idea of being downstream of an ABSENCE of another event (the Notch signal) is a bit slippery. If having expressed the ligand, or receiving another signal, is not necessarily required, one might alternately describe this scenario as the progenitors retaining a pre-existing MCC fate unless they are diverted to the secretory fate through Notch activation. Of note, published data suggest that cells diverted to the secretory fate can be transdifferentiated back to the MCC fate by inhibiting Notch, indicating that the potential to become an MCC is not eliminated, but rather is merely suppressed, by ongoing Notch signaling. Because the opportunity to divert to the secretory cell fate exists prior to MCC differentiation (see below), we choose to retain the language existing in the literature of MCC fate occurring ‘downstream’ of Notch signaling, while expressing this caveat in the amended text.

The second key point is that the entire morphogenetic program of MCC differentiation, including not only centriologenesis, but all the subsequent events including docking, axonemogenesis, and likely numerous other events, are driven by the activation of a MCC-specific transcriptional complex comprised of E2F4/5, Mcidas and DP1 (the EMD complex) to initiate and drive motile ciliogenesis. EMD turns on multiple other secondary transcription factors (including Myb, Rfx3 and Foxj1) and they cooperate to turn on hundreds of structural and regulatory ciliary genes. This leads to activation of the motile cilium biogenesis pathway, which consists of centriole amplification followed by centriole apical transport and membrane docking and then axoneme elongation. These transcription factors are expressed throughout differentiation, and most (with the exception of Foxj1 and possibly Foxn4) are downregulated only after differentiation is complete. Therefore, activating the EMD complex, a step requiring Cdk2, drives not just centriologenesis, but the entirety of the differentiation program. Ciliogenesis is a linear pathway, and a failure to initiate centriole assembly will absolutely preclude subsequent steps in the ciliogenesis program. This in itself does not distinguish between an initiation and a maintenance function for Cdk2. However, our result showing that cells in which Cdk2 is inhibited after the centriole replication has begun nonetheless are arrested in differentiation (Figure 3C) demonstrates an ongoing requirement for Cdk2 in addition to a role in starting the process. This ongoing requirement may reflect a continuing transcriptional activity, but might also, or in addition, be associated with a more direct role of Cdk2 in a downstream process, a possibility suggested by the centriolar pool, for example. We suggest that investigation of the distinct pools of Cdk2 and roles other than initiating and sustaining MCC TF activity are beyond the scope of this manuscript.

With this conceptual framework in mind, we can address some specific queries:

Is Cdk2 directing specifically the initiation and (perhaps) maintenance of MCC centriologenesis or is there a broader role in the (co-)regulation of other steps of cilia assembly.

Our data support a role for Cdk2 in nascent MCCs (in the absence of, or ‘downstream of,’ the Notch signaling event) upstream of the EMD transcriptional complex. Because EMD drives the whole constellation of differentiation events, we propose that Cdk2 is required in nascent MCCs to initiate ciliogenesis. Our data noted above (Figure 3C) also supports an ongoing role for Cdk2 during differentiation.

However, the apparent later stage role of Ccna1, after centriologenesis, seems contradictory to the proposal that Cdk2-Ccna1 are regulating centriole replication and needs to be resolved.

We do not view these roles as contradictory. Both centriole replication and subsequent events are dependent on EMD activation. As described above, both functions might be mediated by Cdk2-Ccna1 acting on EMD, or the later maintenance of ciliogenesis could potentially be mediated by non-transcriptional functions instead of, or in addition to, a transcriptional function.

They need to clarify effects on cell fate specification and ciliogenesis and ciliated cell maintenance.

We have added new data, requested by the reviewers, that incidentally allows us to address this question more precisely. The new data (new Figures 2B and 7C) regards the role of Cdk2 in relation to TRRAP in this process. TRRAP, a histone deacetylase complex member (Wang et al., 2018) was identified as an early regulator of motile ciliogenesis and was shown to act in the nucleus to turn on the EMD complex downstream of (in the absence of) the Notch signaling event. We show that TRRAP expression is induced in the absence of Cdk activity, indicating that TRAPP functions either upstream of, or in parallel to, Cdk2 activation (see new Figure 7C). We allowed MTEC cultures to differentiate for four days in the presence of the Cdk inhibitor Nu6140. As shown before, these cultures are not able to initiate EMD activity and undergo motile ciliogenesis. However Nu6140 treatment had no effect on the cells’ ability to turn on nuclear TRRAP expression. Thus, we while Cdk2 is required to initiate ciliogenesis, we can conclude it is not involved in the MCC fate decision as evidenced by the expression of TRRAP even in the absence of Cdk2 activity.

There might be some experimental limitations that cannot fully separate the roles of Cdk2 in regulating ciliated cell formation and centriologenesis.

We presume here that the reviewer means distinguishing initiation of the differentiation program from a specific and direct function in centriologenesis. Indeed, answering this question would require biochemical analyses to complement the genetic manipulations performed thus far. As described in our response to point 2 below, the asynchrony of differentiation and the limiting amounts of material render these types of studies infeasible. In case we are interpreting the comment incorrectly, and the reviewer means separating the roles of Cdk2 in MCC cell fate specification versus centriologenesis, we argue that the TRRAP result shows that Cdk2 is not required for acquisition of the MCC cell fate.

2) The links between the celluar localization of Cdk2, the activation of Cdk2, and the expression of MCC TFs are inconclusive. It is difficult to appreciate the nuclear-cytoplasmic shuttling during ciliogenesis. In Figure 4, nuclear vs cytoplasmic lysates would be better than this IF (which isn't convincing for nuclear exclusion). One could immunoprecipitate tagged Cdk2 from nuclear and cytoplasmic compartments at various timepoints and show Ccna1 blots?

Two ideas appear to be embedded in this comment. First, “The links between the cellular localization of Cdk2, the activation of Cdk2, and the expression of MCC TFs are inconclusive.” An abundance of prior data establishes that Cdk2 activation and its nuclear localization (and phosphorylation) are coincident. We have shown that its activity is required for EMD complex (MCC TF) activation, and we demonstrate the correlated nuclear localization and Thr160 phosphorylation of Cdk2. These conclusions are based on inhibitor and dominant negative studies, IF assays of nuclear localization (newly quantitated, as described below) and immunoblotting. Thus, the requirement for Cdk2 to activate EMD, and the markers of active Cdk2 are conclusive.

We acknowledge that we do not demonstrate that the EMD complex (MCC TF) is a direct target of Cdk2, and that we provide only immunofluorescence evidence that Cdk2 (and Ccna1) (co)localize in the nucleus in ciliating MCCs. Technical limitations and the nature of MCC differentiation prevent us from addressing these questions biochemically. MTECs (as well as the in vivo airways) undergo MCC fate acquisition and motile ciliogenesis in a highly asynchronous manner (Figure 3). Although younger MTECs are more enriched for cells in the early stages of ciliogenesis and older MTECs contain mostly mature MCCs, MTECs at all times contain MCCs at different levels. Moreover, MTECs contain a large fraction of nonMCCs, which include basal stem cells and secretory cells (see new Figure 1—figure supplement 1C). Our data suggest that Cdk2-Ccna1 is required for ciliogenesis, but not in mature MCCs, and for *Ccna1* we specifically show that its expression is restricted to MCCs. This means that any biochemical fractionation or IP would require purification of MCCs in the process of ciliating from a complex cell population. We have the ability to FACS sort MCCs vs. nonMCCs using the *Foxj1-EGFP* mouse line (Vladar and Stearns, 2007), but this preferentially isolates mature MCCs and cannot distinguish between MCCs at different stages of ciliogenesis (see Figure 1—figure supplement 1 and Figure 3). There are currently no FACS markers specifically for ciliating MCCs or specific ciliogenesis stages. Finally, even if we could sort the desired cells, it is currently not technically feasible to scale up the MTEC culture (limited by the number of basal stem cells isolated from airway tissues) to produce a sufficient quantity of cell lysates for the required fractionation and IP experiments. Experiments from unsorted MTECs would not address these questions or would be impossible to interpret due to the presence of nonMCCs or mature MCCs. It is already well-known from work in other cells and tissues that Cdk2 can undergo nucleo-cytoplasmic shuttling and that it can interact with Ccna1. To add additional value to our manuscript, testing this specifically in ciliating MCCs would be required, but we are unable to isolate these cells.

“It is difficult to appreciate the nuclear-cytoplasmic shuttling during ciliogenesis.” As we are left with only IF assays, we strengthened our observation of nuclear localization by providing careful quantitation of our Cdk2-HA localization in MTECs, the results of which support the presence of Cdk2 in the nucleus in ciliating, but not in mature MCCs (new Figure 4B).

In the manuscript, we are careful to temper our overall conclusions in this revision in light of these limitations. We hope it will be possible to address these points more fully as technology advances. However, the remarkable conservation of Cdk2 localization and function (nuclear localization, partnering with an A-type cyclin and acting on an E2F family TF containing transcriptional complex) between MCCs and cycling cells complements our data and strongly supports our proposed mechanism.

3) The experimental evidence is not sufficient to support that CdK2 acts downstream of Notch signaling. The authors cannot exclude the possibility that Cdk2 and Notch signaling function parallel but share the same downstream consequence.

As discussed above, it is absence of Notch signaling that allows MCC differentiation. To reword the reviewer’s idea, s/he is asking whether Cdk2 might function in parallel and have common downstream consequences with the absence of Notch signaling. Said this way, the idea makes little sense. To reiterate, we argue that progenitors must both NOT have Notch activation, and have activated Cdk2. Since Notch activation suppresses MCC differentiation, temporally the Notch mediated event must occur prior to Cdk2 activation or no secretory cells would be produced. This idea is reinforced by the observation that TRRAP expression, an indicator of entering the MCC pathway (and therefore of having NOT been activated by Notch), occurs in the absence of Cdk2 activity (new Figure 2B). Thus, Notch signaling occurs before Cdk2 activation temporally, but strictly speaking, it is incorrect to say that absence of Notch signaling is upstream of Cdk2 activation.

The other challenging notion in this comment is that there is a downstream consequence of not being activated by Notch. As far as we know, the consequence might be to retain the same state that the cell was in prior to or in the absence of any Notch signaling at all.

Requirements for revision:1) The authors should provide additional evidence that Cdk2 acts downstream of Notch signaling, rather than in a parallel pathway. A further assessment of the components of Notch signaling might be one approach.

We hope that the conceptual challenge in the idea of an activity being downstream of the absence of another signal has been adequately explained above. Strictly speaking, our data are consistent with a model in which both the absence of the Notch signal and the activation of Cdk2 are required in a given cell to promote the MCC fate. Because diversion to the secretory cell fate by Notch activity suppresses Cdk2 activation and the MCC fate, the decision about entering the secretory pathway (active Notch signaling) vs the MCC fate (absence of Notch signaling) must typically occur prior to activation of MCC TFs, and this (together with a lack of complete rigor) is probably the origin of the ‘upstream’ and ‘downstream’ descriptors currently in the literature. We have attempted to convey this admittedly somewhat subtle concept both here and in the revised text (see also the new Figure 1—figure supplement 1A-B), but have chosen to keep the language of ‘upstream’ and ‘downstream’ as shorthand.

Because it is absence of Notch signaling that is required for the MCC cell fate, investigating components of Notch signaling such as Notch intracellular domain (N-ICD) nuclear translocation and resulting gene expression would be relevant to the secretory cell rather than the MCC, and therefore not useful. We attempt to clarify the role of Notch signaling in MCC specification in the text as well as in the new Figure 1—figure supplement 1A-B.

2) The authors should provide more convincing evidence of nuclear-cytoplasmic shuttling with the HA-Cdk2, in control, basal cells and in cells undergoing centriologenesis and ciliogenesis. Identification of the fusion protein in Isolated nuclear and cytoplasmic fractions could be performed.

As stated above, technical limitations and the nature of MCC differentiation prevent us from addressing Cdk2-HA nuclear vs. cytoplasmic enrichment using nuclear and cytoplasmic fraction lysates. Like the in vivo airway surface, MTECs contain MCCs and nonMCCs, and MCCs ciliate asynchronously; thus a bulk extract would contain nuclear and cytoplasmic fractions from cells that would be expected to show multiple Cdk2 enrichment patterns (ciliating vs. mature cells) and thus make it impossible to interpret the results. As described above, we do not have the ability to isolate ciliating MCCs specifically due to the lack of appropriate FACS markers and the low number of these cells in our small scale primary cultures. To support our findings, we now provide quantitation of our Cdk2-HA subcellular localization in MTECs, and the results support the nuclear enrichment of Cdk2 in ciliating, but not in mature MCCs (new Figure 4B).

3) Biochemical evidence of the interaction of Cdk2 with Ccna2 in cyotoplasmic fractions during specific stages of centriologenesis and/or cilia assembly. Related to this strengthen evidence regarding the relationship of HA-Cdk2 and components of centrioles (biochemically or by IF).

This experiment requires demonstration that Cdk2 interacts with Ccna1 (we assume that the Reviewer/Editor is asking about Ccna1 and not Ccna2) specifically in ciliating MCCs. Cdk2 is already known to physically interact with Ccna1 based on work in other systems. However, the technical limitations preclude isolation of this sub-population of cells. We raise and discuss this limitation to our study in the discussion. Our hypothesized mechanism involving a Cdk2-Ccna1 complex in early ciliogenesis regulation is bolstered by analogy to a vast trove of exiting data from other studies that indicate that Cdk2 requires A or E-type cyclin binding for activation.

4) A better characterization of pT160 Cdk2 to identify which specific stages of MCC differentiation are associated with this activity. For example, is there coincident biochemical evidence of high levels of the massive centriole replication (and the presence of other markers of centrioliogenesis) or it later, post centriole replication/docking and instead coincident with cilia assembly? Or, during both processes?

Again, technical limitations do not permit the isolation of the specific cells necessary for a biochemical approach to this question. It might have been possible to address this using IF if the T160 Cdk2 antibody were effective in IF experiments. Unfortunately, we were not able to obtain any immunofluorescence signal with the Cdk2 antibody using multiple fixation methods, suggesting either that the antibody does not work in IF experiments or it is not sufficiently sensitive for the amounts of phospho-Cdk2 expressed. Therefore, we are limited to inference from the increased pT160 Cdk2 signal in early ALI timepoints from bulk isolates that are strongly enriched for MCCs at the earliest stages of ciliogenesis that active Cdk2 is more abundant in these cells.

5) There are several relatively minor points that ought to be addressed related to clarifying the use of specific parameters in graphs, providing statistics, controls for western blots, and quantifying ciliated vs non-ciliated cells in Ccna1 ko mice.

Please note: on 5/2/18 we requested and received the following clarifications on this point by email from Maria Guerreiro, Journal Development Editor*, eLife:*

Concerning Figure 1C/Figure 4—figure supplement 1, the label on the y-axis is not described in the figure legend and is vague. Is confluence of cells binary or quantified? Is this instead confluence/density of MCC? It doesn't change after ALI is established, so how the denominator changes during different stages is unclear. Please clarify.

MTECs proliferate to confluence after about 3-4 days of submerged culture. Confluence is evaluated either visually (no membrane, only uninterrupted epithelial cell surface with cobblestone morphology visible) or it can be inferred from increased transepithelial electrical resistance as measured by an epithelial voltohmmeter. We changed the y-axis labels on both figures to ALI-1d (one day before ALI). The text indicates that this corresponds to a confluent, but not yet differentiating culture. We use cDNA from this time point as a comparison for MCC-related gene expression at later time points as it reflects an intact epithelium with no MCCs.

Figure 5A: the mRNA expression level of Ccna1 at the late ciliogenesis stage is higher than that at the early ciliogenesis stage. But the protein expression levels of Ccna1 in the Figure 5B show an opposite result.

We speculate that at the conclusion of ciliogenesis Ccna1 protein may be actively degraded. This is supported by our inability to detect it by immunofluorescence in individual mature MCCs (Figure 6B) and by analogy with data from other systems and processes that demonstrate that A-type cyclin levels are under strict control by gene expression as well as proteasomal degradation. MTEC lysates and cDNA preps from the three timepoints contain material from a complex mix of MCCs and nonMCCs, although the MCC fraction is enriched for cells at the early, mid and late stages of ciliogenesis. Late ciliogenesis samples also contain many mature and nearly mature MCCs, and it is possible that active degradation of Ccna1 in these cells results in the overall lower signal on the Western blot compared to the corresponding qPCR experiment. It is also possible that the cells that were used for protein lysate and cDNA prep contained a somewhat different proportion of MCCs vs. nonMCCs or ciliating (with higher Ccna1 gene and protein levels) vs. mature (with lower Ccna1 gene and protein levels) MCCs. In either case, both the qPCR and the Western blot results support a trend indicating that actively ciliating MCCs have high Ccna1 transcript and protein levels, which is downregulated as ciliogenesis concludes – supporting a role for Ccna1 in regulating this process.

In addition to quantifying ciliated vs. non-ciliated cells in Ccna1ko mice and including Western blot loading controls, there are two other minor things that need to be clarified: 1. Figure 3: The authors stated that they did not detect a statistically significant difference in the Myb+/Foxj1- population under the Cdki treatment. However, in the Figure 3C, the fraction of Myb+/Foxj1- is significantly decreased (p<0.05) in the Nu6140 acute treated ALI at D4 as compared with untreated ALI at D3.

Quantification of the *Ccna1KO* results is now included in the manuscript (new Figure 7—figure supplement 7A).

Ponceau-S stained membranes serving as loading control for Western blots are now show in the following: new Figure 4—figure supplement 4B for Figure 4C and new Figure 5—figure supplement 5B for Figure 5B.

We thank the Reviewers for catching this, as it is an error. We have now corrected the figure to indicate that there was no significant change detected in the Myb+/Foxj1- population in the Nu6140 acute treated ALI+4d as compared with untreated ALI+3d (t=0).

6) Finally, the last four paragraphs of the Discussion could be condensed since it is speculation and not directly focused on the data. I think that one major emphasis of the discussion should be a comparison to Al Jord et al., 2017, who show that Cdk1 (rather than Cdk2) in balance with APC is required for centriole amplification-ciliation and inhibition of mitosis, respectively. The Vlader discussion should further explain why Al Jord implicates Cdk1 and Vlader Cdk2 beyond what is written in paragraph two. The authors should discuss specifically if they believe that their data indicate that Cdk2 is directly a program or has specific protein targets these issues are discussed loosely in paragraphs three, four and five. Consider inhibition of APC and examine Cdk2-dependent centriole amplification, this would address the discussion point in paragraph six.

The Discussion has been condensed and we now discuss and compare in more detail our study regarding the role of Cdk2 and the Al Jord study on Cdk1. The APC inhibition experiment is interesting but we feel that it is beyond the scope of this study.